# ADAM EXPLOITS $\ell_\infty$-GEOMETRY OF LOSS LANDSCAPE VIA COORDINATE-WISE ADAPTIVITY

**Shuo Xie, Mohamad Amin Mohamadi & Zhiyuan Li**
Toyota Technological Institute at Chicago
{shuox,mohamadamin,zhiyuanli}@ttic.edu

## ABSTRACT

Adam outperforms SGD when training language models. Yet this advantage is not well-understood theoretically – previous convergence analysis for Adam and SGD mainly focuses on the number of steps $T$ and is already minimax-optimal in non-convex cases, which are both $\widetilde{O}(T^{-1/4})$. In this work, we argue that the exploitation of nice $\ell_\infty$-geometry is the key advantage of Adam over SGD. More specifically, we give a new convergence analysis for Adam under novel assumptions that loss is smooth under $\ell_\infty$-geometry rather than the more common $\ell_2$-geometry, which yields a much better empirical smoothness constant for GPT-2 and ResNet models. Our experiments confirm that Adam performs much worse when the favorable $\ell_\infty$-geometry is changed while SGD provably remains unaffected. We also extend the convergence analysis to blockwise Adam under novel blockwise smoothness assumptions.

## 1 INTRODUCTION

Large language models (LLMs) have gained phenomenal capabilities as their scale grows (Kaplan et al., 2020; Brown et al., 2020; Touvron et al., 2023; OpenAI, 2023; Reid et al., 2024). However, pre-training LLMs is incredibly time-consuming. Adam (Kingma & Ba, 2014) is the current to-go optimization algorithm for LLMs due to its fast convergence. In contrast, SGD, a popular and arguably the simplest optimizer, optimizes language model losses much more slowly than Adam.

However, the optimization benefit of Adam over SGD cannot be explained by existing theory. Current convergence analyses for Adam and SGD focus on the dependence on the number of steps under assumptions on loss smoothness and gradient bounds (Défossez et al., 2022), and it has been shown that both Adam and SGD achieve the minimax convergence rate $\widetilde{O}(T^{-1/4})$ in the non-convex settings (Arjevani et al., 2023). Thus according to the theory, in the worst case, SGD would be more desirable than Adam because it has the same convergence rate, and yet Adam is less memory-efficient due to its coordinate-wise adaptivity, which needs to store the empirical moving average of second-order moments of past stochastic gradients. Therefore, we hypothesize that the coordinate-wise adaptivity in Adam is exploiting some unknown properties of LLMs which SGD cannot make use of.

Towards this end, we identified a big difference between Adam and SGD, which is ignored in previous works. That is, SGD is rotation-equivariant, while Adam is only permutation equivariant (Definition 2.1). Intuitively, if we rotate the loss landscape, the trajectory of SGD would be the same (up to some rotation), while the trajectory of Adam can be completely different. If Adam optimizes much more slowly after rotation, it suggests Adam is exploiting some non-rotation-invariant properties of the loss, which is not captured by standard smoothness assumptions in the convergence analysis.

Figure 1 summarizes our findings by comparing Adam on the original and rotated loss. The performance of Adam on the rotated loss does become much worse than Adam on the original loss. We also test a memory-efficient and rotation-equivariant variant of SGD, AdaSGD (Wang & Wiens, 2020), defined in Algorithm 2. Surprisingly, the rotated Adam performs even worse than the SGD variant. The results suggest it is impossible to explain the superior optimization performance of Adam over SGD just using rotation-invariant assumptions on the loss function, which raises the natural question,

*What non-rotation-invariant properties of loss functions make* Adam *converge faster than* SGD*?*

We hypothesize that the $\ell_2$-lipschitzness of loss gradient does not provide a tight-enough characterization of loss landscape of deep learning models in practice, such that we can separate Adam and

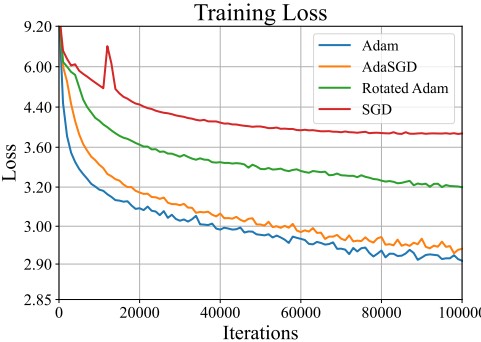 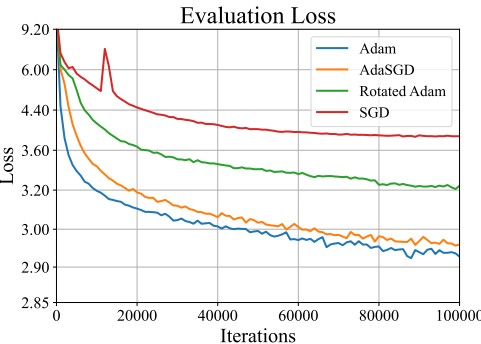

Figure 1: Training and evaluation losses of `Adam`, `AdaSGD` and `SGD` on GPT-2. `rotated Adam` means running `Adam` on a rotated loss. `Adam` on the original loss converges the fastest as expected. But convergence of `Adam` on a rotated loss is much slower, notably even worse than `AdaSGD`.

other rotation-equivariant algorithms. Inspired by the similarity between `Adam` and `SignGD` and the fact that `SignGD` is the normalized steepest descent with respect to $\ell_\infty$-norm, we propose to use $\ell_\infty$-norm related smoothness as a better tool to analyze `Adam`. In particular, our main results use the $(1,1)$-norm of the Hessian of the loss normalized by variable dimension $d$, as the smoothness measure, instead of its spectral norm. And we prove a convergence rate of $O(\frac{1}{\sqrt{T}})$ for `Adam` without noise, or $O((\frac{\log T}{T})^{1/4})$ with noise. Our results have the same dependence on $T$ as previous results, but a much smaller smoothness constant when measured empirically. We empirically verify that $(1,1)$-norm of Hessian positively correlates with final training loss of `Adam` on both synthetic tasks like quadratic loss and real tasks like training GPT2 on OpenWebText and ResNet on CIFAR10.

We summarize our contributions below:

1. We show by experiments that the empirical optimization advantage of `Adam` over `SGD` can not be explained solely under rotation-invariant assumptions. (Figure 1)

2. We propose a new complexity metric for the optimization problem, which is the $(1,1)$-norm of the Hessian matrix of loss, $\left\|\nabla^2 L(x)\right\|_{1,1}$. We present a novel convergence result for `Adam` depending on this metric in the case of $\beta_1 = 0$. (Theorem 3.5 )

3. We further generalize the theoretical analysis (Theorem 3.11) for `Adam` to blockwise `Adam` (Algorithm 3) whose convergence rate can be characterized by a novel smoothness measure (Definition 3.9). `Adam` and `AdaSGD` are two notable examples of blockwise `Adam`. In `Adam`, all blocks are of size $1$. In `AdaSGD`, there is only one block.

4. We empirically verify that when `Adam` converges more slowly on the rotated loss, the $(1,1)$-norm of Hessian also increases, which suggests that our new complexity metric for `Adam`'s convergence is practically relevant. (Section 4).[1]

## 2 PRELIMINARIES

**Notations.** For $\boldsymbol{x} \in \mathbb{R}^d$, we define the vector $p$-norm $\|\boldsymbol{x}\|_p$ as $(\sum_{i=1}^d x_i^p)^{1/p}$ for $p \in [1, \infty]$. For a matrix $\boldsymbol{A} \in \mathbb{R}^{d_1 \times d_2}$, its $(1,1)$-norm $\|\boldsymbol{A}\|_{1,1}$ is defined as $\sum_{i=1}^{d_1} \sum_{j=1}^{d_2} |A_{i,j}|$ and its operator norm induced by vector $p$-norm $\|\cdot\|_p$ as $\sup_{\boldsymbol{x} \in \mathbb{R}^d} \frac{\|\boldsymbol{A}\boldsymbol{x}\|_q}{\|\boldsymbol{x}\|_p}$, denoted by $\|\boldsymbol{A}\|_p$, where $\frac{1}{q} + \frac{1}{p} = 1$ and $\|\cdot\|_q$ is the dual norm of $\|\cdot\|_p$. For a square matrix $\boldsymbol{A} \in \mathbb{R}^{d \times d}$, $|\boldsymbol{A}|$ is defined as the unique square root of $\boldsymbol{A}^\top \boldsymbol{A}$. For a deterministic loss function $L(\boldsymbol{x})$, we consider optimization over $L$ with access only to independent stochastic functions $\{L_t(\boldsymbol{x})\}_{t=1}^T$ such that $\mathbb{E}L_t(\boldsymbol{x}) = L(\boldsymbol{x})$ for any input $\boldsymbol{x} \in \mathbb{R}^d$.

**Rotation.** For an invertible function $\mathcal{T} : \mathbb{R}^d \to \mathbb{R}^d$, $\mathcal{T}$ is a rotating transformation if there exists an orthogonal matrix $\boldsymbol{T} \in \mathbb{R}^{d \times d}$ such that $\mathcal{T}(\boldsymbol{x}) = \boldsymbol{T}\boldsymbol{x}$. $\mathcal{T}$ is a permutating transformation if there exists a permutation $\pi : [d] \to [d]$ such that $\mathcal{T}(\boldsymbol{x}) = [x_{\pi(1)}, \ldots, x_{\pi(d)}]^\top$. A permutating transformation is always a rotating transformation. We will use $\mathcal{R}$ to denote a rotating transformation.

**Definition 2.1.** *For initialization $\boldsymbol{x}_0$ and stochastic losses $\{L_t\}_{t=1}^T$, we can get $\boldsymbol{x}_t$ when running algorithm $A$ on $(\boldsymbol{x}_0, \{L_t\}_{t=1}^T)$. For a transformation $\mathcal{T}$, we can also get $\tilde{\boldsymbol{x}}_t$ when running $A$ with the same hyperparameters on $(\tilde{\boldsymbol{x}}_0, \{\tilde{L}_t\}_{t=1}^T)$ with $\tilde{\boldsymbol{x}}_0 = \mathcal{T}^{-1}(\boldsymbol{x}_0)$ and $\tilde{L}_t = L_t \circ \mathcal{T}$.*

---

[1]The code is at `https://github.com/mohamad-amin/adam-coordinate-adaptivity`.

---

**Algorithm 1** Adam

**Hyperparam:** $\beta_1, \beta_2, \epsilon \geq 0$, total steps $T$, learning rate $\{\eta_t\}_{t=1}^T$, initial $\boldsymbol{m}_0, v_0$
**Input:** initial $\boldsymbol{x}_0$, stochastic losses $\{L_t\}_{t=1}^T$

  $v_{0,i} \leftarrow v_0$
  **for** $t = 1, 2, \cdots, T$:
    $g_{t,i} \leftarrow \nabla_i L_t(\boldsymbol{x}_{t-1})$
    $m_{t,i} \leftarrow \beta_1 m_{t-1,i} + (1-\beta_1)g_{t,i}$
    $v_{t,i} \leftarrow \beta_2 v_{t-1,i} + (1-\beta_2)g_{t,i}^2$
    $x_{t,i} \leftarrow x_{t-1,i} - \eta_t \frac{m_{t,i}}{\sqrt{v_{t,i}+\epsilon}}$
  **return** $\boldsymbol{x}_T$

---

**Algorithm 2** AdaSGD (Wang & Wiens, 2020)

**Hyperparam:** $\beta_1, \beta_2, \epsilon \geq 0$, total steps $T$, learning rate $\{\eta_t\}_{t=1}^T$, initial $\boldsymbol{m}_0, v_0$
**Input:** initial $\boldsymbol{x}_0$, stochastic losses $\{L_t\}_{t=1}^T$

  **for** $t = 1, 2, \cdots, T$:
    $g_{t,i} \leftarrow \nabla_i L_t(\boldsymbol{x}_{t-1})$
    $m_{t,i} \leftarrow \beta_1 m_{t-1,i} + (1-\beta_1)g_{t,i}{}^2$
    $v_t \leftarrow \beta_2 v_{t-1} + (1-\beta_2)(\|\boldsymbol{g}_t\|_2^2 /d)$
    $x_{t,i} \leftarrow x_{t-1,i} - \eta_t \frac{m_{t,i}}{\sqrt{v_t+\epsilon}}$
  **return** $\boldsymbol{x}_T$

---

*An algorithm A is* equivariant w.r.t. $\mathcal{T}$ *if it always holds that $\tilde{\boldsymbol{x}}_t = \mathcal{T}^{-1}(\boldsymbol{x}_t)$ for any hyperparameters, initialization and stochastic losses. A is* rotation-equivariant *if it is equivariant w.r.t. any rotating transformation $\mathcal{R}$. A is* permutation-equivariant *if it is equivariant w.r.t. any permutating transformation.*

Theorem 2.2 shows the difference between `Adam` and `AdaSGD`, whose proof is in Appendix B.

**Theorem 2.2.** `AdaSGD` *is rotation-equivariant.* `Adam` *and* `SignGD` *are only permutation-equivariant.*

# 3 MAIN RESULTS: CONVERGENCE RATES OF Adam

In this section, we present our main theoretical results. Theorem 3.5 gives a convergence analysis of `Adam` for stochastic non-convex smooth losses with coordinate-wise gradient noise. The convergence is measured by $\ell_1$ norm of the gradient. For deterministic losses, our best rate (Theorem 3.2) is achieved by `SignGD` (`Adam` with $\beta_1 = \beta_2 = 0$). For stochastic losses with bounded gradient noise variance, our best rate (Corollary 3.6) is achieved by `RMSProp` (`Adam` with $\beta_1 = 0, \beta_2 \in [0,1]$).

Then we extend our analysis of `Adam` to more general blockwise `Adam` (Theorem 3.11), which contains both `Adam` and `AdaSGD` as special cases. We also come up with novel smoothness measures (Definition D.2) corresponding to the set of blocks used in blockwise `Adam`.

Similar to previous works (Défossez et al., 2022), our analysis can be extended to the general case of `Adam`, where both $\beta_1, \beta_2$ are non-zero. But the rate becomes strictly worse than the `RMSProp` (the case of $\beta_1 = 0$), as there will be some extra polynomials of $\frac{1}{1-\beta_1}$. We decide not to include the result for the general case, on the one hand for ease of presentation, and on the other hand, because such result can not explain the optimization benefit of momentum ($\beta_1 > 0$) in practice and does not add any insight on the benefit of `Adam`. We hypothesize that the theoretical assumptions are missing some important features of loss landscape of transformers and we leave this for future work.

## 3.1 WARMUP: SignGD (Adam WITH $\beta_1 = \beta_2 = 0$)

In this section, we present the convergence analysis for `SignGD` as a warm-up and illustrate how `Adam` could benefit from a non-rotation-invariant property of the loss, which in particular is the $\ell_\infty$ smoothness. The key observation is that `SignGD` is normalized steepest descent w.r.t. $\ell_\infty$ norm (Xie & Li, 2024), so it is more natural to analyze its convergence using $\ell_\infty$ norm geometry of the loss.

**Definition 3.1.** *Given a norm $\|\cdot\|$ on $\mathbb{R}^d$ and $\|\cdot\|_*$ as its dual norm, we say a function $L$ is $H$-smooth w.r.t. $\|\cdot\|$ if for any $\boldsymbol{x}, \boldsymbol{y} \in \mathbb{R}^d$, we have that $\|\nabla L(\boldsymbol{x}) - \nabla L(\boldsymbol{y})\|_* \leq H \|\boldsymbol{x} - \boldsymbol{y}\|$.*

**Theorem 3.2.** *Let $L$ be $H$-smooth w.r.t. $\ell_\infty$ norm and $\{\boldsymbol{x}_t\}_{t=1}^T$ be the iterates of* `SignGD` *(`Adam` with $\beta_1 = \beta_2 = 0$) on $L$ with initialization $\boldsymbol{x}_0$ and learning rate $\eta$, it holds that*

$$\min_{1 \leq t \leq T} \|\nabla L(\boldsymbol{x}_t)\|_1 \leq \frac{L(\boldsymbol{x}_0) - \min L(\boldsymbol{x})}{T\eta} + \frac{H\eta}{2}.$$

*If we choose $\eta = \sqrt{\frac{2(L(\boldsymbol{x}_0)-\min L(\boldsymbol{x}))}{TH}}$, then $\min_{1 \leq t \leq T} \|\nabla L(\boldsymbol{x}_t)\|_1 \leq \sqrt{\frac{2H(L(\boldsymbol{x}_0)-\min L(\boldsymbol{x}))}{T}}$.*

---

[2]Here it is slightly different from Wang & Wiens (2020). We use an exponential average of the gradient for $\boldsymbol{m}_t$ instead of momentum. Our definition makes `AdaSGD` the same as `Adam` in a one-dimensional problem.

Theorem 3.2 gives the convergence rate for `SignGD` and the proof is in Appendix C.

### 3.2 MAIN RESULT: RMSProp (`Adam` WITH $\beta_1 = 0, \beta_2 \in [0, 1]$)

It is well-known that `SignGD` might not converge in the stochastic case as the expectation of descent direction for mini-batch loss may not be a descent direction for $L$. `RMSProp` is proposed to address this issue by using a moving average of the squared gradient per coordinate to reduce the correlation between the denominator and the numerator, thus making the expected update direction less biased (Hinton et al., 2012). In this subsection we formalize the above intuition and show indeed a positive $\beta_2$ in `Adam` helps convergence in the stochastic case. The main challenges here are from both lower bounding the first-order term and upper bounding the second-order term in the modified descent lemma (the `RMSProp` counterpart of Equation 7).

$$L(\boldsymbol{x}_t) - L(\boldsymbol{x}_{t-1}) \leq -\eta_t \nabla L(\boldsymbol{x}_t)^\top \frac{\boldsymbol{g}_t}{\sqrt{\boldsymbol{v}_t + \epsilon}} + \frac{H}{2}\eta_t^2 \left\| \frac{\boldsymbol{g}_t}{\sqrt{\boldsymbol{v}_t + \epsilon}} \right\|_\infty^2$$

We can only upper bound $\left\| \frac{\boldsymbol{g}_t}{\sqrt{\boldsymbol{v}_t + \epsilon}} \right\|_\infty^2$ by $\frac{1}{1 - \beta_2}$ without more fine-grained analysis on the relationship between gradients in each step, which will greatly hurt the dependence of convergence rate on $1 - \beta_2$. However, even though some large $g_{t,i}$ can make $\frac{g_{t,i}}{\sqrt{v_{t,i} + \epsilon}}$ as large as $\frac{1}{\sqrt{1 - \beta_2}}$, the average coordinate moving speed should be close to 1. Therefore, we introduce Definition 3.3, which is slightly stronger than Definition 3.1 but enables decomposing the second order term into each coordinate according to Lemma D.3. It also facilitates coordinate-wise analysis for the first order term. We note this definition also appears in Assumption 2.3 of the concurrent work (Maladkar et al., 2024).

**Definition 3.3.** *For any* $\mathbf{H} = (H_1, \ldots, H_d) \in \mathbb{R}^d$, *we say $L$ is $\mathbf{H}$-smooth coordinate-wisely w.r.t. $\ell_\infty$ norm , if and only if* $|\nabla_i L(\boldsymbol{x}) - \nabla_i L(\boldsymbol{y})| \leq H_i \|\boldsymbol{x} - \boldsymbol{y}\|_\infty$ *for any* $i \in [d]$, $\boldsymbol{x}, \boldsymbol{y} \in \mathbb{R}^d$.

**(1,1)-norm as an estimate for coordinate-wise smoothness.** $H_i$ in Definition 3.3 is determined by $\sup_{\boldsymbol{x}} \sum_{j=1}^d |\nabla_{i,j}^2 L(\boldsymbol{x})|$, which is difficult to compute because it requires taking supreme over the entire domain. A computationally-tractable alternative is to approximate $\sum_{i=1}^d H_i$ locally by the $(1, 1)$-norm of Hessian of loss along the training trajectory. We provide an efficient approximation algorithm with concentration guarantees in Appendix E.3, which uses hessian-vector product against random Cauchy vectors.

By definition, $\mathbf{H}$-smoothness coordinate-wisely w.r.t. $\ell_\infty$ norm implies $\sum_{i=1}^d H_i$-smoothness w.r.t. $\ell_\infty$ norm. We also need Assumption 3.4 to measure the influence of noise in the stochastic setting.

**Assumption 3.4** (Coordinate-wise noise)**.** *There exist constants $\sigma_i$ such that* $\mathbb{E}\left[\nabla_i L_t(\boldsymbol{x}) - \nabla_i L(\boldsymbol{x})\right]^2 \leq \sigma_i^2$ *for any* $i \in [d]$, $t \in \mathbb{N}$ *and* $\boldsymbol{x} \in \mathbb{R}^d$.

We present the main result in Theorem 3.5. The sketch of the proof is presented in Section 3.4 and the complete proof for the generalized blockwise Adam algorithm is presented in Appendix D.1. The proof incorporates some key steps from Li & Lin (2024), extending them to accommodate the generalized algorithm and different smoothness assumptions.

**Theorem 3.5** (Main, `Adam`)**.** *Let $\{L_t\}_{t=1}^T$ be independent stochastic losses satisfying Assumption 3.4 and that their expectation $L$ is $\mathbf{H}$-coordinate-wisely smooth w.r.t. $\ell_\infty$ norm. For `Adam` with $\beta_1 = 0$, we have that*

$$\min_{\frac{T}{2} < t \leq T} \mathbb{E} \|\nabla L(\boldsymbol{x}_t)\|_1 \leq O\left( E + \sqrt{E}\sqrt{\frac{\beta_2^{\frac{T}{4}}}{T(1 - \beta_2)} dv_0 + \sum_{i=1}^d \sigma_i + d\sqrt{\epsilon}} \right)$$

*with* $E = \frac{2}{\eta T}\mathbb{E}\left[L(\boldsymbol{x}_0) - L(\boldsymbol{x}_T)\right] + \left(1 + \frac{\beta_2 F}{T(1 - \beta_2)}\right)\left(\eta \sum_{i=1}^d H_i + \sqrt{1 - \beta_2} \sum_{i=1}^d \sigma_i\right)$ *and* $F = 2\ln\left(1 + \frac{\sum_{i=1}^d \sigma_i^2 + \|\nabla L(\boldsymbol{x}_0)\|_\infty^2 + \sum_{i \in [d]} H_i^2 \eta^2 T(T + \frac{1}{1 - \beta_2})}{v_0 + \epsilon}\right) + \ln 32$.

The convergence rate of RMSprop can be determined by choosing hyperparameters $\eta$ and $\beta_2$ in Theorem 3.5 to minimize $E$. By assuming $v_0 + \epsilon > (\sum_{i=1}^d \sigma_i^2 + \|\nabla L(\boldsymbol{x}_0)\|_\infty^2 + \sum_i H_i^2 \eta^2)/\texttt{poly}(T)$ and

$\frac{1}{1-\beta_2} = \texttt{poly}(T)$, we can simplify the term by considering $F = O(\log T)$. The two terms involving $\sum_{i=1}^{d} \sigma_i$ have a lower bound $\Theta\left(\sum_{i=1}^{d} \sigma_i \left(\frac{\log T}{T}\right)^{\frac{1}{2}}\right)$, which is achieved with $1 - \beta_2 = \Theta\left(\frac{\log T}{T}\right)$. Then the three terms involving $\eta$ has a lower bound $\Theta\left(\sqrt{\frac{(L(\boldsymbol{x}_0) - \min_{\boldsymbol{x}} L(\boldsymbol{x})) \sum_{i=1}^{d} H_i}{T}}\right)$ reached by $\eta = \Theta\left(\sqrt{\frac{L(\boldsymbol{x}_0) - \min_{\boldsymbol{x}} L(\boldsymbol{x})}{T \sum_{i=1}^{d} H_i}}\right)$. These hyperparameter choices yield the optimal convergence rate for stochastic case in Corollary 3.6. For convenience, we define $R \triangleq (L(\boldsymbol{x}_0) - \min_{\boldsymbol{x}} L(\boldsymbol{x})) \sum_{i=1}^{d} H_i$, which will be the core term in Corollaries 3.6 and 3.7.

**Corollary 3.6** (Stochastic Case, general $\sigma_i$). *Let $\{L_t\}_{t=1}^{T}$ be independent stochastic losses satisfying Assumption 3.4 and that their expectation $L$ is $\boldsymbol{H}$-coordinate-wisely smooth w.r.t. $\ell_\infty$ norm. For $\beta_1 = 0$, $1 - \beta_2 = \Theta(\frac{\log T}{T})$, $\epsilon = 0$, $\eta = \Theta\left(\sqrt{\frac{L(\boldsymbol{x}_0) - \min_{\boldsymbol{x}} L(\boldsymbol{x})}{T \sum_{i=1}^{d} H_i}}\right)$ and $v_0 > (\sum_{i=1}^{d} \sigma_i^2 + \|\nabla L(\boldsymbol{x}_0)\|_\infty^2 + \sum_i H_i^2 \eta^2)/\texttt{poly}(T)$, we have that*

$$\min_{\frac{T}{2} < t \leq T} \mathbb{E} \|\boldsymbol{g}_t\|_1 = O\left(\sqrt{\frac{R}{T}} + \sqrt{\sum_{i=1}^{d} \sigma_i \left(\frac{R}{T}\right)^{\frac{1}{4}}} + \sum_{i=1}^{d} \sigma_i \left(\frac{\log T}{T}\right)^{\frac{1}{4}} + \delta_T\right)$$

*with $\delta_T = \sqrt{\frac{dv_0}{T(1-\beta_2)}} \exp\left(-\frac{T(1-\beta_2)}{8}\right) \left[\left(\frac{R}{T}\right)^{\frac{1}{4}} + \sqrt{\sum_{i=1}^{d} \sigma_i \left(\frac{\log T}{T}\right)^{\frac{1}{4}}}\right]$.*

Here $\delta_T$ can be smaller than any polynomial of $T$ by manipulating the value of $\frac{T(1-\beta_2)}{\log T} = \Theta(1)$. Then $\sum_{i=1}^{d} \sigma_i \left(\frac{\log T}{T}\right)^{\frac{1}{4}}$ is the leading term w.r.t. $T$ in the rate whose coefficient only involves $\sum_{i=1}^{d} \sigma_i$. It suggests that the rate can be much improved when noise is small. Below we get the convergence rate with the same hyperparameters in deterministic case in Corollary 3.7.

**Corollary 3.7** (Deterministic Case, $\sigma_i = 0$). *Let $\{L_t\}_{t=1}^{T}$ be deterministic losses satisfying Assumption 3.4 and that their expectation $L$ is $\boldsymbol{H}$-coordinate-wisely smooth w.r.t. $\ell_\infty$ norm. For $\beta_1 = 0$, $1 - \beta_2 = \Omega(\frac{\log T}{T})$, $\epsilon = 0$, $\eta = \Theta\left(\sqrt{\frac{L(\boldsymbol{x}_0) - \min_{\boldsymbol{x}} L(\boldsymbol{x})}{T \sum_{i=1}^{d} H_i}}\right)$ and $v_0 > (\sum_{i=1}^{d} \sigma_i^2 + \|\nabla L(\boldsymbol{x}_0)\|_\infty^2 + \sum_i H_i^2 \eta^2)/\texttt{poly}(T)$ for any polynomial $\texttt{poly}(T)$, we have that*

$$\min_{\frac{T}{2} < t \leq T} \|\boldsymbol{g}_t\|_1 = O\left(\sqrt{\frac{R}{T}} + \delta_T\right)$$

*with $\delta_T = \sqrt{\frac{dv_0}{T(1-\beta_2)}} \exp\left(-\frac{T(1-\beta_2)}{8}\right) \left(\frac{R}{T}\right)^{\frac{1}{4}}$.*

Corollary 3.7 almost recovers Theorem 3.2, except for the smoothness constant. Specifically, it uses $\sup_{\boldsymbol{x}} \|\nabla^2 L(\boldsymbol{x})\|_{(1,1)}$ that is larger than $\sup_{\boldsymbol{x}} \|\nabla^2 L(\boldsymbol{x})\|_\infty$ in Theorem 3.2 as $\|\cdot\|_{1,1} \geq \|\cdot\|_\infty$ always holds. This gap is due to technical difficulty of analyzing $\texttt{Adam}$ as mentioned in Section 3.2.

**Dependence on $\epsilon$, $v_0$ and $\beta_2$.** While many previous works rely on the relatively large magnitude of $\epsilon$ compared to $\boldsymbol{v}_t$ and give a bound in the regime of $\texttt{SGD}$ when the adaptive effect is dominated by the constant $\epsilon$ (Zaheer et al., 2018; De et al., 2018), our result actually prefers $\epsilon$ to be 0 while maintaining the value of $v_0 + \epsilon$. We also note the dependence of our bound in Theorem 3.5 on $v_0$ is very mild and logarithmic. Theorem 3.5 has similar convergence rates for all $v_0$ of magnitude at most $\texttt{poly}(T)$, while most previous result only addresses the case where $v_{0,i}$ is at the scale of noise (Li & Lin, 2024) or 0. The main reason for this adaptivity to a wide range of $v_0$ is our specific choice of $\beta_2 = 1 - \Theta(\frac{\log T}{T})$, which allows the initial large $v_0$ to decay fast and resume normal training. Other existing results using $\beta_2 = 1 - \Theta(1/T)$ (Défossez et al., 2022; Li & Lin, 2024) cannot allow large initial value $v_0$ because $v_0$ only decays a constant fraction throughout the training and the effective learning rate will be too small.

### 3.3 A UNIFIED ANALYSIS FOR BLOCKWISE $\texttt{Adam}$

In this subsection, we present convergence analysis for a broader class of adaptive algorithms defined in Algorithm 3. It can be viewed as a coarser version of $\texttt{Adam}$ because it does pre-conditioning

---

**Algorithm 3** Blockwise `Adam`

---

**Hyperparam:** $\beta_1, \beta_2, \epsilon \geq 0$, block partition $\Phi$, total steps $T$, learning rate $\{\eta_t\}_{t=1}^T$, initial $\boldsymbol{m}_0, v_0$.
**Input:** initial $\boldsymbol{x}_0$, stochastic losses $\{L_t\}_{t=1}^T$

    $v_{0,b} \leftarrow v_0$
    **for** $t = 1, 2, \cdots, T$ :
        $g_{t,i} \leftarrow \nabla_i L_t(\boldsymbol{x}_{t-1})$
        $m_{t,i} \leftarrow \beta_1 m_{t-1,i} + (1 - \beta_1) g_{t,i}$
        $v_{t,b} \leftarrow \beta_2 v_{t-1,b} + (1 - \beta_2) \frac{\sum_{\Phi(i)=b} g_{t,i}^2}{d_b}$
        $x_{t,i} \leftarrow x_{t-1,i} - \eta_t \frac{m_{t,i}}{\sqrt{v_{t,\Phi(i)} + \epsilon}}$
    **return** $\boldsymbol{x}_T$

---

blockwisely instead of coordinate-wisely. Since `Adam` and `AdaSGD` can be viewed as special cases of blockwise `Adam` with $\Phi_{\texttt{Adam}} : i \mapsto i$ and $\Phi_{\texttt{AdaSGD}} : i \mapsto 1$ respectively, any convergence results for Algorithm 3 would imply convergence of `Adam` and `AdaSGD`. Finally we also note that such blockwise `Adam` has been recently studied empirically by some concurrent work, where the algorithm is named by Adam-mini (Zhang et al., 2024b) and Adalayer (Zhao et al., 2024).

We first introduce more notations. For a partition function $\Phi : [d] \to [B]$ where $B$ is the number of blocks, $(b)$ is defined as $\Phi^{-1}(b) = \{i | \Phi(i) = b\}$ and $d_b = \#(b)$, the number of parameters in block $b$. We define the vector $\boldsymbol{x}_{(b)}$ as $[x_i]_{\Phi(i)=b}$ and the submatrix $\boldsymbol{A}_{(b),(b')}$ as $[A_{i,j}]_{\Phi(i)=b, \Phi(j)=b'}$.

**Definition 3.8** ($\Phi$-norm). *We define the $(\infty, 2)$-norm w.r.t. partition $\Phi$ of $\boldsymbol{x}$ as the $\ell_\infty$ norm of the vector* $\left( \frac{\|\boldsymbol{x}_{(b)}\|_2}{\sqrt{d_b}} \right)_{b=1}^B$, *which is* $\max_{b \in [B]} \frac{\|\boldsymbol{x}_{(b)}\|_2}{\sqrt{d_b}}$. *For convenience, we will denote it by* $\|\boldsymbol{x}\|_\Phi$ *or just call it $\Phi$-norm. We denote its dual norm by* $\|\boldsymbol{x}\|_{\Phi,*}$, *which is equal to* $\sum_{b=1}^B \sqrt{d_b} \|\boldsymbol{x}_{(b)}\|_2$.

**Definition 3.9** ($\Phi$-smoothness). *We say a diagonal matrix $\boldsymbol{A}$ follows partition $\Phi$ iff its diagonal elements are constant within each block, i.e., there are $a_1, \cdots, a_B$ s.t. $\boldsymbol{A}_{i,i} = a_{\Phi(i)}$ for any $i \in [d]$.*

*We say a twice-differentiable $L$ is $H$-smooth under partition $\Phi$ if there exists a diagonal matrix $\boldsymbol{A}$ following partition $\Phi$ such that $H = \text{Tr}(\boldsymbol{A})$ and $\boldsymbol{A}$ dominates $|\nabla^2 L(\boldsymbol{x})|$ for all $\boldsymbol{x}$. We further define the $\Phi$-smoothness of loss $L$, denoted by $H(L, \Phi)$, as the smallest constant $H$ such that $L$ is $H$-smooth under parition $\Phi$, that is,*

$$H(L, \Phi) = \min_{\boldsymbol{A} \text{ follows } \Phi, \ \boldsymbol{A} \succeq |\nabla^2 L(\boldsymbol{x})|, \forall \boldsymbol{x}} \text{Tr}(\boldsymbol{A}) \tag{1}$$

We note that $\Phi$-smoothness is different from the smoothness under $\Phi$-norm, where the latter is equal to $\sup_{\boldsymbol{x} \in \mathbb{R}^d} \sup_{\|\boldsymbol{u}\|_\Phi \leq 1} \|\nabla^2 L(\boldsymbol{x})\boldsymbol{u}\|_{\Phi,*}$. For each $\boldsymbol{x}$, it holds that

$$\sup_{\|\boldsymbol{u}\|_\Phi \leq 1} \|\nabla^2 L(\boldsymbol{x})\boldsymbol{u}\|_{\Phi,*} = \sup_{\|\boldsymbol{u}\|_\Phi \leq 1} \sup_{\|\boldsymbol{v}\|_\Phi \leq 1} \boldsymbol{v}^\top \nabla^2 L(\boldsymbol{x})\boldsymbol{u} = \sup_{\|\boldsymbol{u}\|_\Phi \leq 1} |\boldsymbol{u}^\top \nabla^2 L(\boldsymbol{x})\boldsymbol{u}|.$$

When $\boldsymbol{A} \succeq |\nabla^2 L(\boldsymbol{x})|$, we have that $\sup_{\|\boldsymbol{u}\|_\Phi \leq 1} |\boldsymbol{u}^\top \nabla^2 L(\boldsymbol{x})\boldsymbol{u}| \leq \sup_{\|\boldsymbol{u}\|_\Phi \leq 1} \boldsymbol{u}^\top \boldsymbol{A}\boldsymbol{u}$. When diagonal $\boldsymbol{A}$ follows partition $\Phi$, we have that $\sup_{\|\boldsymbol{u}\|_\Phi \leq 1} \boldsymbol{u}^\top \boldsymbol{A}\boldsymbol{u} = \text{Tr}(\boldsymbol{A})$. So $\Phi$-smoothness defined in Definition 3.9 is always no smaller than the smoothness under $\Phi$-norm.

Another advantage with the definition is that the $H(L, \Phi)$ will always non-increase with a more fine-grained partition. For two partition functions $\Phi_1$ and $\Phi_2$, we say $\Phi_1$ includes $\Phi_2$ if and only if $\Phi_1(i) = \Phi_1(j)$ for any $i, j \in [d]$ such that $\Phi_2(i) = \Phi_2(j)$. If a diagonal matrix $\boldsymbol{A}$ follows partition $\Phi_1$ and partition $\Phi_1$ includes $\Phi_2$, then $\boldsymbol{A}$ also follows $\Phi_2$. So we have that $\{\boldsymbol{A} \mid \boldsymbol{A} \text{ follows } \Phi_1, \boldsymbol{A} \succeq |\nabla^2 L(\boldsymbol{x})| \text{ for all } \boldsymbol{x}\} \subseteq \{\boldsymbol{A} \mid \boldsymbol{A} \text{ follows } \Phi_2, \boldsymbol{A} \succeq |\nabla^2 L(\boldsymbol{x})| \text{ for all } \boldsymbol{x}\}$ and $H(L, \Phi_1)$ is no smaller than $H(L, \Phi_2)$. Since $\Phi_{\texttt{AdaSGD}}$ includes any partition and any partition includes $\Phi_{\texttt{Adam}}$, $H(L, \Phi_{\texttt{AdaSGD}})$ is the largest and $H(L, \Phi_{\texttt{Adam}})$ is the smallest among all the partitions $\Phi$.

With Assumption 3.10 on noise, we can prove the result for blockwise `Adam` in Theorem 3.11.

**Assumption 3.10** (Generalized version of Assumption 3.4). *There exists constant $\sigma_b$ such that* $\mathbb{E} \|\nabla_{(b)} L_t(\boldsymbol{x}) - \nabla_{(b)} L(\boldsymbol{x})\|_2^2 \leq d_b \sigma_b^2$ *for any block $b \in [B], t \in \mathbb{N}$ and $\boldsymbol{x} \in \mathbb{R}^d$.*

**Theorem 3.11** (Main, Blockwise `Adam`). *For a specific partition $\Phi$, we consider the updates defined in Algorithm 3. Under Assumption 3.10, we have that*

$$\min_{\frac{T}{2} < t \leq T} \mathbb{E} \sum_{b=1}^B \sqrt{d_b} \|\nabla_{(b)} L(\boldsymbol{x}_t)\|_2 \leq 2\sqrt{2}E + \sqrt{2E} \sqrt{\frac{4\beta_2^{\frac{T}{4}}}{T(1 - \beta_2)} d\sqrt{v_0} + \sum_{b=1}^B d_b \sigma_b + d\sqrt{\epsilon}}$$

*with $E = \frac{2}{\eta T}\mathbb{E}\left[L(\boldsymbol{x}_0) - L(\boldsymbol{x}_T)\right] + \left(1 + \frac{\beta_2 F}{T(1-\beta_2)}\right)\left(\eta H(L, \Phi) + 2\sqrt{1-\beta_2}\sum_{b=1}^{B} d_b \sigma_b\right)$ and $F =$* $2\ln\left(1 + \frac{\sum_{b=1}^{B}\sigma_b^2 + \|\nabla L(\boldsymbol{x}_0)\|_\Phi^2 + \sum_{b\in[B]} H_b^2 d_b \eta^2 T(T + \frac{1}{1-\beta_2})}{v_0 + \epsilon}\right) + \ln 32$.

When $L$ is $(H_1, \ldots, H_d)$-smooth coordinate-wisely w.r.t. $\ell_\infty$ norm, we show in Appendix D.2 that $\operatorname{diag}(H_1, \ldots, H_d) \succeq \left|\nabla^2 L(\boldsymbol{x})\right|$ for all $\boldsymbol{x} \in \mathbb{R}^d$ and $\operatorname{diag}(H_1, \ldots, H_d)$ follows $\Phi_{\texttt{Adam}}$ partition. So $H(L, \Phi_{\texttt{Adam}})$ is at most $\sum_{i=1}^{d} H_i$ and we can use Theorem 3.11 to derive Theorem 3.5. We can also estimate $H(L, \Phi_{\texttt{Adam}})$ by $\sup_{\boldsymbol{x}\in\mathbb{R}^d}\left\|\nabla^2 L(\boldsymbol{x})\right\|_{1,1}$ as discussed in Section 3.2. For $\Phi_{\texttt{AdaSGD}}$ being the mapping $i \mapsto 1$, $H(L, \Phi_{\texttt{AdaSGD}})$ is the same as the smoothness under $\Phi_{\texttt{AdaSGD}}$-norm, whose value is $d\sup_{\boldsymbol{x}\in\mathbb{R}^d}\left\|\nabla^2 L(\boldsymbol{x})\right\|_2$.

**Different norms for smoothness.** As an implication of Theorem 3.11, we can get analogs of Corollaries 3.6 to 3.7 for $\texttt{AdaSGD}$, with the corresponding noise and smoothness assumptions. When the optimization is not noise-dominated and $\sqrt{\frac{R}{T}} = \sqrt{\frac{(L(\boldsymbol{x}_0) - \min_{\boldsymbol{x}} L(\boldsymbol{x}))H(L,\Phi)}{T}}$ becomes the leading term, the choice of $\Phi$ now matters a lot. The key difference between $\texttt{AdaSGD}$ and $\texttt{Adam}$ lies in the gap between $H(L, \Phi_{\texttt{AdaSGD}})$ and $H(L, \Phi_{\texttt{Adam}})$ and comparing these coefficients can provide insight into which algorithm may be more effective under different conditions.

Previous analyses of $\texttt{Adam}$'s convergence (Shi & Li, 2021; Défossez et al., 2022; Li & Lin, 2024) usually assume smoothness under the $\ell_2$ norm and the rate of $\texttt{Adam}$ ends up being identical to the rate of $\texttt{AdaSGD}$, which fails to explain why $\texttt{Adam}$ often performs better than $\texttt{AdaSGD}$ in practice. By adopting an $\ell_\infty$ norm smoothness assumption, the coefficient for $\texttt{Adam}$'s convergence rate changes from $d\sup_{\boldsymbol{x}}\|\nabla^2 L(\boldsymbol{x})\|_2$ to $H(L, \Phi_{\texttt{Adam}})$, where the latter is typically much smaller when $\texttt{Adam}$ optimizes faster because $\sup_{\boldsymbol{x}}\left\|\nabla^2 L(\boldsymbol{x})\right\|_{1,1}$ is much smaller than $d\sup_{\boldsymbol{x}}\left\|\nabla^2 L(\boldsymbol{x})\right\|_2$.

Finally, we note that the $\Phi_{\texttt{Adam}}$-smoothness $H(L, \Phi_{\texttt{Adam}})$ is not rotation-invariant in the sense that $H(L, \Phi_{\texttt{Adam}}) \neq H(L \circ \mathcal{R}, \Phi_{\texttt{Adam}})$ for a typical rotation $\mathcal{R}$. In practice, the $(1,1)$-norm of Hessian matrix can vary a lot when a rotation is performed on the loss as shown in Section 4.1. In contrast, $\Phi_{\texttt{AdaSGD}}$-smoothness $H(L, \Phi_{\texttt{AdaSGD}})$ is invariant under loss rotations.

## 3.4 Proof Sketch of Theorem 3.11

We will use $\bar{\boldsymbol{g}}_t = \mathbb{E}[\boldsymbol{g}_t|\boldsymbol{x}_{<t}] = \nabla L(\boldsymbol{x}_{t-1})$ to denote the full batch gradient and consider the decrease of $L(\boldsymbol{x}_t)$ in a single step $t$. We can upper bound the second order term in the Taylor expansion (Equation 2) with $\boldsymbol{H} \succeq \nabla^2 L(\boldsymbol{x})$ that achieves $\operatorname{Tr}(\boldsymbol{H}) = H(L, \Phi)$. Then we can get

$$L(\boldsymbol{x}_t) - L(\boldsymbol{x}_{t-1}) \leq -\eta\sum_{i=1}^{d}\frac{g_{t,i}\bar{g}_{t,i}}{\sqrt{v_{t,\Phi(i)} + \epsilon}} + \frac{1}{2}\eta^2\sum_{b=1}^{B} H_b \frac{\left\|\boldsymbol{g}_{t,(b)}\right\|_2^2}{v_{t,b} + \epsilon} \tag{2}$$

$$= -\eta\sum_{b=1}^{B}\frac{\boldsymbol{g}_{t,(b)}^\top \bar{\boldsymbol{g}}_{t,(b)}}{\sqrt{v_{t,b} + \epsilon}} + \frac{1}{2}\eta^2\sum_{b=1}^{B} H_b d_b \frac{\left\|\boldsymbol{g}_{t,(b)}\right\|_2^2 / d_b}{v_{t,b} + \epsilon} \tag{3}$$

The proof contains two main parts: lower bounding the first order term using $\|\bar{\boldsymbol{g}}_t\|_{\Phi,*}$ and upper bounding the second order term. We address the second-order term by employing Lemma 3.12 to bound the sum by $T + \frac{\beta_2}{1-\beta_2}\ln\frac{v_{T,b}+\epsilon}{v_{0,b}+\epsilon}$, where we set $v_t \triangleq v_{t,b}$ and $g_t \triangleq \left\|\boldsymbol{g}_{t,(b)}\right\|_2/\sqrt{d_b}$.

**Lemma 3.12.** *Given any $0 < \beta_2 < 1$, for any scalar sequences $\{v_t\}_{t=0}^{T}$ and $\{g_t\}_{t=1}^{T}$ satisfying $v_0 \geq 0, v_1 > 0$ and $v_t - \beta_2 v_{t-1} \geq (1-\beta_2)g_t^2$ for $t \geq 1$, it holds that $\sum_{t=1}^{T}\frac{g_t^2}{v_t} \leq T + \frac{\beta_2}{1-\beta_2}\ln\frac{v_T}{v_0}$.*

Now we turn to the first term. Ideally, for each block $b \in [B]$, we would like to connect the first order term to $\|\bar{\boldsymbol{g}}_{t,(b)}\|_2^2$ by taking expectation, i.e., $\mathbb{E}_t\frac{\boldsymbol{g}_{t,(b)}^\top \bar{\boldsymbol{g}}_{t,(b)}}{\sqrt{v_{t,b}+\epsilon}} \approx \frac{\mathbb{E}_t \boldsymbol{g}_{t,(b)}^\top \bar{\boldsymbol{g}}_{t,(b)}}{\sqrt{v_{t,b}+\epsilon}} = \frac{\|\bar{\boldsymbol{g}}_{t,(b)}\|_2^2}{\sqrt{v_{t,b}+\epsilon}}$, where we use $\mathbb{E}_t[\cdot]$ as abbreviation for $\mathbb{E}[\cdot|\boldsymbol{x}_{<t}]$. However, this is not correct because both the numerator and denominator in $\frac{\boldsymbol{g}_{t,(b)}^\top \bar{\boldsymbol{g}}_{t,(b)}}{\sqrt{v_{t,b}+\epsilon}}$ depend on the stochastic gradient $\boldsymbol{g}_t$. To circumvent this difficulty, we lower bound each conditional expectation $\mathbb{E}_t\frac{\boldsymbol{g}_{t,(b)}^\top \bar{\boldsymbol{g}}_{t,(b)}}{\sqrt{v_{t,b}+\epsilon}}$ by $\frac{\mathbb{E}_t \boldsymbol{g}_{t,(b)}^\top \bar{\boldsymbol{g}}_{t,(b)}}{2\sqrt{\mathbb{E}_t v_{t,b}+\epsilon}}$, minus error

terms related to noise magnitude $\sigma_b$. We can further have $\frac{\mathbb{E}_t \boldsymbol{g}_{t,(b)}^\top \bar{\boldsymbol{g}}_{t,(b)}}{\sqrt{\mathbb{E}_t v_{t,b}+\epsilon}} \geq \frac{\|\bar{\boldsymbol{g}}_{t,(b)}\|_2^2}{\sqrt{\tilde{v}_{t,b}+\epsilon}}$, where $\tilde{v}_{t,b} =$
$\beta_2 v_{t-1,b} + (1-\beta_2)\left(\|\bar{\boldsymbol{g}}_{t,(b)}\|_2^2 / d_b + \sigma_b^2\right)$. This leads to Lemma 3.13 whose proof is in Appendix D.

**Lemma 3.13** (first-order approximation). *With Assumption 3.10, it holds that for any block $b \in [B]$*

$$\mathbb{E}\sum_{t=1}^T \frac{\boldsymbol{g}_{t,(b)}^\top \bar{\boldsymbol{g}}_{t,(b)}}{\sqrt{v_{t,b}+\epsilon}} \geq \frac{1}{2}\mathbb{E}\sum_{t=1}^T \frac{\|\bar{\boldsymbol{g}}_{t,(b)}\|_2^2}{\sqrt{\tilde{v}_{t,b}+\epsilon}} - \sqrt{1-\beta_2}Td_b\sigma_b - \frac{d_b\sigma_b\beta_2}{\sqrt{1-\beta_2}}\mathbb{E}\left[\ln\frac{v_{T,b}+\epsilon}{v_{0,b}+\epsilon}\right]. \quad (4)$$

Combining Lemmas 3.12 and 3.13 and Equation 3 gives an upper bound for $\mathbb{E}\sum_{t=1}^T \sum_{b=1}^B \frac{\|\bar{\boldsymbol{g}}_{t,(b)}\|_2^2}{\sqrt{\tilde{v}_{t,b}+\epsilon}}$.
Finally, we employ Cauchy inequality (Equation 5) and Lemma 3.14 to upper bound $\|\bar{\boldsymbol{g}}_t\|_{\Phi,*}$ using
$\mathbb{E}\sum_{t=\frac{T}{2}+1}^T \sum_{b=1}^B \frac{\|\bar{\boldsymbol{g}}_{t,(b)}\|_2^2}{\sqrt{\tilde{v}_{t,b}+\epsilon}}$. This completes the proof.

$$\sum_{t=\frac{T}{2}+1}^T \|\bar{\boldsymbol{g}}_t\|_{\Phi,*} = \sum_{t=\frac{T}{2}+1}^T \sum_{b=1}^B \sqrt{d_b}\|\bar{\boldsymbol{g}}_{t,(b)}\|_2 \leq \sqrt{\sum_{t=\frac{T}{2}+1}^T \sum_{b=1}^B \frac{\|\bar{\boldsymbol{g}}_{t,(b)}\|_2^2}{\sqrt{\tilde{v}_{t,b}+\epsilon}}} \sqrt{\sum_{t=\frac{T}{2}+1}^T \sum_{b=1}^B d_b\sqrt{\tilde{v}_{t,b}+\epsilon}}. \quad (5)$$

**Lemma 3.14.** *With Assumption 3.10, it holds that for any block $b \in [B]$*

$$\sum_{t=\frac{T}{2}+1}^T \mathbb{E}\left[\sqrt{\tilde{v}_{t,b}+\epsilon}\right] \leq \frac{2\beta_2^{\frac{T}{4}}}{1-\beta_2}\sqrt{v_{0,b}} + \frac{T}{2}\sigma_b + \frac{T}{2}\sqrt{\epsilon} + 2\sum_{t=1}^T \mathbb{E}\left[\frac{\|\bar{\boldsymbol{g}}_{t,(b)}\|_2^2 / d_b}{\sqrt{\tilde{v}_{t,b}+\epsilon}}\right]. \quad (6)$$

## 4 EXPERIMENTS

In order to empirically investigate and confirm the implications of our proposed theory, we compare the training performance of `Adam` with `AdaSGD`, `SGD` and `rotated Adam` on multiple different tasks. The details of all experiments are in Appendix E.1.

### 4.1 QUADRATIC LOSS

We perform controlled experiments on quadratic loss to study the relationship between optimization speed of `Adam` and the shape of Hessian in terms of $\Phi_{\text{Adam}}$-smoothness. More specifically, we consider $\Sigma = \text{diag}(1, \cdots, 1, \frac{1}{2^2}, \frac{1}{3^2}, \cdots, \frac{1}{990^2}) \in \mathbb{R}^{1000 \times 1000}$ and manually generate orthogonal matrices $\mathcal{R}_i$. Then we optimize $L_0(\boldsymbol{x}) = \frac{1}{2}\boldsymbol{x}^\top \Sigma \boldsymbol{x}$ with `AdaSGD` and `Adam` and optimize $L_i(\boldsymbol{x}) = \frac{1}{2}\boldsymbol{x}^\top \mathcal{R}_i^\top \Sigma \mathcal{R}_i \boldsymbol{x}$ with `Adam` for 100 steps. Because `AdaSGD` is rotation-equivariant, the optimization process of `AdaSGD` is the same on all $L_i$. We tune learning rates for each setting with 10 random seeds and present their lowest average loss with standard deviation in Table 1.

| Optimizer | $(1,1)$-**norm**$/d$ | Loss ($\beta_1 = \beta_2 = 0$) | Loss ($\beta_1 = 0.9, \beta_2 = 0.99$) |
|---|---|---|---|
| AdaSGD | 0.01164 | $0.00887 \pm 0.00119$ | $0.00405 \pm 0.00021$ |
| Adam | 0.01164 | $0.00022 \pm 0.00007$ | $0.00002 \pm 0.00001$ |
| Adam ($\mathcal{R}_1$) | 0.08324 | $0.00314 \pm 0.00031$ | $0.00066 \pm 0.00008$ |
| Adam ($\mathcal{R}_2$) | 0.50729 | $0.00567 \pm 0.00053$ | $0.00134 \pm 0.00007$ |
| Adam ($\mathcal{R}_3$) | 1.23731 | $0.00751 \pm 0.00086$ | $0.00183 \pm 0.00009$ |
| Adam ($\mathcal{R}_4$) | 2.59919 | $0.00978 \pm 0.00132$ | $0.00254 \pm 0.00008$ |

Table 1: The final loss values obtained by different optimizers and the $(1,1)$-norm of Hessian matrix for the corresponding objectives. The spectral norm of the Hessian is always 1. `Adam` optimizes worse when the $(1,1)$-norm of Hessian matrix increases, as suggested by Corollary 3.7. Moreover, when $(1,1)$-norm divided by $d$ is much smaller than spectral norm, `Adam` tends to optimize faster than `AdaSGD`, supporting $\Phi$-smoothness as a predictor of blockwise `Adam`'s optimization speed.

We find a clear pattern that `Adam` optimizes worse when the $(1,1)$-norm of Hessian matrix increases, as suggested by our Corollary 3.7. Moreover, when $(1,1)$-norm divided by dimension is smaller than spectral norm, `Adam` tends to optimize faster than `AdaSGD`, as suggested by our Theorem 3.11.

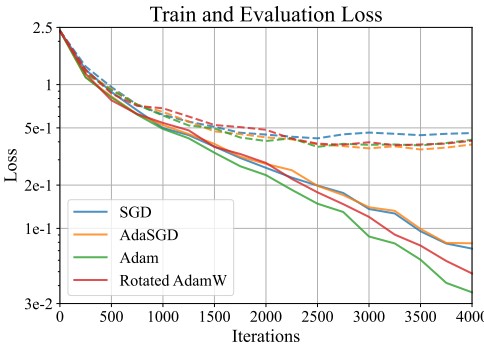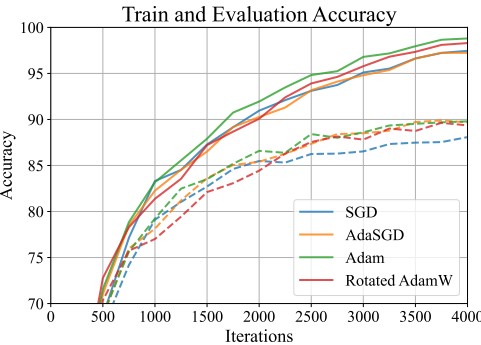

Figure 2: Training (solid line) and test (dashed line) losses and accuracy of ResNet18 on CIFAR-10 with `Adam`, `AdaSGD`, `rotated Adam`, and `SGD`. `Adam` converges faster than other algorithms.

## 4.2 GPT-2 ON LANGUAGE MODELING TASK

We train GPT-2 on the OpenWebText corpus. The training losses and evaluation losses of different optimizers are plotted in Figure 1. As mentioned in Section 1, `Adam` converges faster than `AdaSGD` while they both converge faster than `rotated Adam`. Since we propose the $(1,1)$-norm of Hessian as a non-rotation-invariant metric that can affect the convergence rate of `Adam`, we also measure it for the original loss function $L$ and rotated loss function $\tilde{L}$ on checkpoints trained with different losses. The results are presented in Table 2. The same correlation between norms and convergence rates holds here. The smaller the norm is, the faster the optimizer works.

| Optimizer | Smoothness Metric | Upper Bound | Estimated Value |
|-----------|-------------------|-------------|-----------------|
| AdaSGD | $H(L, \Phi_{\texttt{AdaSGD}})$ | $d \left\lVert \nabla^2 L(\boldsymbol{x}) \right\rVert_2$ | 4.2446 |
| Adam | $H(L, \Phi_{\texttt{Adam}})$ | $\left\lVert \nabla^2 L(\boldsymbol{x}) \right\rVert_{1,1}$ | 2.3538 |
| Rotated Adam | $H(L \circ \mathcal{R}, \Phi_{\texttt{Adam}})$ | $\left\lVert R^{\top} \nabla^2 L(\boldsymbol{x}) R \right\rVert_{1,1}$ | 14.3745 |

Table 2: Hessian norms for the last GPT-2 checkpoints trained with different optimizers.

We also explore how learning rate can affect the performance of different optimizers and find another advantage of `Adam` over `AdaSGD`: it can maintain stable training at a larger learning rate, which is often beneficial to faster and more efficient convergence. The results and details are in Appendix E.4.

GPT-2 small models have more than 100 million parameters, and thus the size of its hessian of loss as well as the rotation matrix is more than $10^{16}$, which is way more than the storage of the modern computers. We introduce how to rotate the loss efficiently in Appendix E.2 and how to estimate Hessian norms in Appendix E.3.

## 4.3 RESNET18 ON CIFAR-10

To further test whether the correlation between $\Phi$-smoothness and the optimization performance holds for architectures other than transformers, we conduct an experiment on ResNet18 trained on CIFAR-10 (Krizhevsky & Hinton, 2009).

Figure 2 depicts the loss and accuracy curves for the best performing hyperparameters chosen over the training set's final loss for batch size 256.[3] The results for other batch sizes are in Table 4. When it comes to optimization speed, even for ResNet18, `Adam` is always better than `rotated Adam` and they are always better than `AdaSGD` and `SGD` across different batch sizes. Note that this does not contradict with common practice of training ResNet with `SGD`, where the goal is to get better generalization and the training budget is large so all optimizers can easily achieve full training accuracy. In our experiment, we study optimization speed and intentionally limit the number of steps.

---

[3]We have intentionally limited the number of training iterations to emphasize the difference of optimizers in terms of training speed over generalization.

| Optimizer | Smoothness Metric | Upper Bound | Estimated Value |
|-----------|-------------------|-------------|-----------------|
| AdaSGD | $H(L, \Phi_{\mathtt{AdaSGD}})$ | $d \left\| \nabla^2 L(\boldsymbol{x}) \right\|_2$ | 1.5355 |
| Adam | $H(L, \Phi_{\mathtt{Adam}})$ | $\left\| \nabla^2 L(\boldsymbol{x}) \right\|_{1,1}$ | 0.0036 |
| Rotated Adam | $H(L \circ \mathcal{R}, \Phi_{\mathtt{Adam}})$ | $\left\| R^\top \nabla^2 L(\boldsymbol{x}) R \right\|_{1,1}$ | 0.9868 |

Table 3: Hessian norms for optimal ResNet checkpoints trained with different optimizers.

We also measure the Hessian for checkpoints obtained at batch size 256 and the results are in Table 3. The correlation between norms and convergence rates still holds here. When the $(1,1)$-norm is smaller than $d$ times spectral norm, Adam optimizes faster than AdaSGD.

## 5 RELATED WORKS

**Comparison between Adam and SGD**  Previous work tries to analyze the difference between Adam and SGD from different perspectives. Zhou et al. (2018) proves a faster convergence rate of Adam than SGD when the stochastic gradients are sparse. Zhang et al. (2020) suggests that SGD suffers more from heavy-tailed noise than Adam. Pan & Li (2023) claims that Adam has lower directional sharpness because of the effect of coordinate-wise clipping. Other works also consider the coordinate-wise normalization of Adam (Balles & Hennig, 2018; Kunstner et al., 2022). Kunstner et al. (2024) shows that the heavy-tailed class imbalance in language modeling tasks will cause SGD to converge slower when it can only optimize majority class well. Zhang et al. (2024a) finds that Adam is better at handling the block heterogeneity of Hessian matrix, which is a specific phenomenon in transformers. When viewing Adam as an adaptive method, there are works showing that adaptive methods have an advantage of achieving optimal convergence rate without relying on problem-dependent constant (Ward et al., 2020; Levy et al., 2021).

**Convergence rate of Adam**  There are many works showing convergence rate for Adam (Zhou et al., 2018; Chen et al., 2018; Zou et al., 2019; Shi & Li, 2021; Guo et al., 2021; Défossez et al., 2022; Zhang et al., 2022). Most of them rely on the smoothness of the loss function, which is measured w.r.t. $\ell_2$ norm. Zhang et al. (2019) proposes $(L_0, L_1)$-smoothness condition should be more reasonable than globally bounded smoothness. Li et al. (2024) further generalizes the $(L_0, L_1)$ smoothness condition. However, they still focus on the default rotation-invariant $\ell_2$ norm. To the best of our knowledge, we are the first to assume smoothness under $\ell_\infty$ norm for analyzing Adam.

**Comparison with Li & Lin (2024)**  Li & Lin (2024) employs the same $\ell_1$ norm for gradient and improves the dependence on dimension $d$ compared to previous results for $\ell_2$ norm. But they still assume $\ell_2$ norm smoothness while we adapt their results under $\ell_\infty$ norm smoothness to potentially further improve dependence on $d$. Another drawback of Li & Lin (2024) is setting $\boldsymbol{v}_0$ based on noise magnitude $\sigma$. which is impractical in real experiments because $\sigma$ is unknown. Overestimating $\sigma$ will result in slow convergence because large $\boldsymbol{v}_0$ causes Adam to behave similarly with SGD without adjusting the learning rate adaptively. In contrast, we allow for general initialization for $\boldsymbol{v}_0$ and our convergence rate can work well in both noisy setting and deterministic setting. We also use $1 - \beta_2 = \Theta\left(\frac{\log T}{T}\right)$ to obtain our convergence rate while Li & Lin (2024) requires $1 - \beta_2 = \Theta\left(\frac{1}{T}\right)$.

## 6 CONCLUSION

We give a new convergence analysis (Theorem 3.5) for Adam in the stochastic non-convex setting using a novel smoothness assumption. We show the convergence rate for the $\ell_1$ norm of the gradient is $O\left(\frac{1}{\sqrt{T}}\right)$ in the deterministic case (Corollary 3.7) and $O\left(\left(\frac{\log T}{T}\right)^{\frac{1}{4}}\right)$ in the stochastic case (Corollary 3.6). We also extend our analysis to blockwise Adam on loss $L$ with respect to an arbitrary partition of the parameters $\Phi$ (Theorem 3.11) using the corresponding smoothness $H(L, \Phi)$ (Definition D.2). Our bound for Adam involves $(1,1)$-norm of Hessian, rather than the spectral norm of Hessian, which is relevant to the convergence speed of AdaSGD. This leads to significantly better smoothness conditions for deep learning models including ResNet-18 and GPT2 empirically. Our experiments also verify that the smoothness measure $H(L, \Phi)$ positively correlates with the optimization speed of blockwise Adam with respect to the partition $\Phi$.

ACKNOWLEDGEMENT

The authors would like to thank Khashayar Gatmiry and Sharut Gupta for helpful discussion and preliminary experiments in exploring the idea of rotated Adam and Xiaoyu Chen for discussions on extending the analysis to blockwise Adam and steepest descent w.r.t. general $\Phi$-norm. ZL is supported by OpenAI Superalignment Grant.

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

CONTENTS

## A   MORE RELATED WORK

We note the definition of $\Phi_{\texttt{Adam}}$-smoothness been mentioned in previous work. Bernstein et al. (2018) relies on it to prove the convergence rate of SignGD. Previous and concurrent work Ene et al. (2021); Liu et al. (2024); Jiang et al. (2024) uses this assumption for analyzing AdaGrad. We are the first to analyze Adam under such smoothness assumption. Moreover, we are the first to empirically measure it for loss functions in real task with theoretical guarantee. Concurrent work Maes et al. (2024) proposes another algorithm to measure $(1, 1)$-norm of Hessian matrix. Ling et al. (2022) analyzes the rotation equivariance property of Adam in the geometry optimization setting.

## B   INVARIANCE PROPERTY OF Adam AND SGD

**Theorem 2.2.** Adam *is rotation-equivariant.* Adam *and* SignGD *are only permutation-equivariant.*

*Proof of Theorem 2.2.* For SGD and AdaSGD, we will show they are rotation-equivariant by induction. For any rotating transformation $\mathcal{R}(\boldsymbol{x}) = \boldsymbol{R}\boldsymbol{x}$, suppose $\tilde{\boldsymbol{x}}_s = \mathcal{R}^{-1}(\boldsymbol{x}_s) = \boldsymbol{R}^\top \boldsymbol{x}_s$ holds for $s \leq t - 1$. Then we have that $\tilde{\boldsymbol{g}}_t = \nabla_{\tilde{\boldsymbol{x}}} \tilde{L}_t(\tilde{\boldsymbol{x}}_t) = \boldsymbol{R}^\top \nabla_{\boldsymbol{x}} L(\boldsymbol{R}^{-1}\tilde{\boldsymbol{x}}_{t-1}) = \boldsymbol{R}^\top \nabla_{\boldsymbol{x}} L(\boldsymbol{x}_{t-1}) = \boldsymbol{R}^\top \boldsymbol{g}_t$ and $\tilde{\boldsymbol{m}}_t = \boldsymbol{R}^\top \boldsymbol{m}_t$. From the update rule of SGD, we have that $\tilde{\boldsymbol{x}}_t = \tilde{\boldsymbol{x}}_{t-1} - \eta_t \tilde{\boldsymbol{m}}_t = \boldsymbol{R}^\top \boldsymbol{x}_{t-1} - \eta_t \boldsymbol{R}^\top \boldsymbol{m}_t = \boldsymbol{R}^\top (\boldsymbol{x}_{t-1} - \eta_t \boldsymbol{m}_t) = \boldsymbol{R}^\top \boldsymbol{x}_t$. For the update rule of AdaSGD, we further have that $\|\tilde{\boldsymbol{g}}_t\|_2^2 = \|\boldsymbol{g}_t\|_2^2$ because $\boldsymbol{R}$ is an orthogonal matrix. Then $\tilde{v}_t = v_t$ and the derivation is similar.

For Adam and SignGD, it is easy to show by induction they are equivariant w.r.t. any permutating transformation because the operation on gradient is performed on each coordinate separately. We only need to show they are not equivariant w.r.t. a rotating transformation. We choose $\boldsymbol{R} = [\frac{1}{\sqrt{2}}, \frac{1}{\sqrt{2}}; \frac{1}{\sqrt{2}}, -\frac{1}{\sqrt{2}}]$, $L_t(\boldsymbol{x}) = L(\boldsymbol{x}) = 2x_1^2 + x_2^2$. Due to the update rule of SignGD, it can only update $\boldsymbol{x}$ and $\tilde{\boldsymbol{x}}$ in the direction of $[1, 1]$ and $[1, -1]$. But when rotating the update direction on $\tilde{\boldsymbol{x}}$ back to the space of $\boldsymbol{x}$. The update direction can only be $[1, 0]$ or $[0, 1]$ that are different from the update direction in the original space. Because the first step in Adam takes the same direction in SignGD, we simultaneously show that both SignGD and Adam are not rotation-equivariant. $\square$

## C   CONVERGENCE RATE OF SignGD FOR DETERMINISTIC LOSS

**Theorem 3.2.** *Let $L$ be $H$-smooth w.r.t. $\ell_\infty$ norm and $\{\boldsymbol{x}_t\}_{t=1}^T$ be the iterates of* SignGD *(*Adam *with $\beta_1 = \beta_2 = 0$) on $L$ with initialization $\boldsymbol{x}_0$ and learning rate $\eta$, it holds that*

$$\min_{1 \leq t \leq T} \|\nabla L(\boldsymbol{x}_t)\|_1 \leq \frac{L(\boldsymbol{x}_0) - \min L(\boldsymbol{x})}{T\eta} + \frac{H\eta}{2}.$$

*If we choose $\eta = \sqrt{\frac{2(L(\boldsymbol{x}_0) - \min L(\boldsymbol{x}))}{TH}}$, then $\min_{1 \leq t \leq T} \|\nabla L(\boldsymbol{x}_t)\|_1 \leq \sqrt{\frac{2H(L(\boldsymbol{x}_0) - \min L(\boldsymbol{x}))}{T}}$.*

*Proof of Theorem 3.2.* We will directly prove a more general verion of Theorem 3.2. Because $L$ is $H$-smooth with respect to $\|\cdot\|_\infty$, we have that

$$L(\boldsymbol{x}_{t+1}) - L(\boldsymbol{x}_t) \leq -\nabla L(\boldsymbol{x}_t)^\top (\boldsymbol{x}_t - \boldsymbol{x}_{t+1}) + \frac{H}{2} \|\boldsymbol{x}_t - \boldsymbol{x}_{t+1}\|^2$$

$$\leq -\eta \|\nabla L(\boldsymbol{x}_t)\|_* + \frac{\eta^2 H}{2}\eta^2 \tag{7}$$

This implies that

$$\min_{1 \leq t \leq T} \|\nabla L(\boldsymbol{x}_t)\|_* \leq \frac{1}{T} \sum_{t=1}^T \|\nabla L(\boldsymbol{x}_t)\|_* \leq \frac{L(\boldsymbol{x}_0) - L(\boldsymbol{x}_T)}{T\eta} + \frac{H\eta}{2},$$

which completes the proof. $\square$

## D   PROOF DETAILS

### D.1   PROOF FOR CONVERGENCE RATE OF BLOCKWISE Adam

We will use Lemma 3.12 to better control the growth of the sum of second order term.

**Lemma 3.12.** *Given any $0 < \beta_2 < 1$, for any scalar sequences $\{v_t\}_{t=0}^T$ and $\{g_t\}_{t=1}^T$ satisfying $v_0 \geq 0, v_1 > 0$ and $v_t - \beta_2 v_{t-1} \geq (1 - \beta_2)g_t^2$ for $t \geq 1$, it holds that $\sum_{t=1}^T \frac{g_t^2}{v_t} \leq T + \frac{\beta_2}{1-\beta_2} \ln \frac{v_T}{v_0}$.*

*Proof of Lemma 3.12.* Notice that $1 - x \leq \ln \frac{1}{x}$ for any positive $x$. We can have that

$$
\begin{aligned}
\sum_{t=1}^{T} \frac{g_t^2}{v_t} &\leq \sum_{t=1}^{T} \frac{v_t - \beta_2 v_{t-1}}{(1 - \beta_2) v_t} \\
&= \sum_{t=1}^{T} \left[ 1 + \frac{\beta_2}{1 - \beta_2} \left( 1 - \frac{v_{t-1}}{v_t} \right) \right] \\
&\leq T + \frac{\beta_2}{1 - \beta_2} \sum_{t=1}^{T} \ln \frac{v_t}{v_{t-1}} \\
&= T + \frac{\beta_2}{1 - \beta_2} \ln \frac{v_T}{v_0}.
\end{aligned}
\tag{8}
$$

when $v_0 \neq 0$. When $v_0 = 0$, we can still have that

$$
\begin{aligned}
\sum_{t=1}^{T} \frac{g_t^2}{v_t} &\leq \frac{1}{1 - \beta_2} + \sum_{t=2}^{T} \frac{g_t^2}{v_t} \\
&\leq \frac{1}{1 - \beta_2} + (T - 1) + \frac{\beta_2}{1 - \beta_2} \ln \frac{v_T}{v_1} \\
&= T + \frac{\beta_2}{1 - \beta_2} \ln \frac{v_T}{v_1/e}.
\end{aligned}
$$

$\square$

As mentioned in Section 3.4, we deal with the first order term by approximating it with a deterministic term. Recall the notation defined in Section 3.4. $g_t$ denotes the gradient of mini-batch $L_t(x_{t-1})$ at step $t$. And $\mathbb{E}[g_t | x_{t-1}] = \nabla L(x_{t-1})$ because $\mathbb{E} L_t = L$. The full-batch gradient is $\bar{g}_t = \nabla L(x_{t-1})$. Different kinds of second-order momentum are defined in the following way

$$
v_{t,b} = \beta_2^t \left\| g_{1,(b)} \right\|_2^2 / d_b + (1 - \beta_2) \sum_{j=0}^{t-1} \beta_2^j \left( \left\| g_{t-j,(b)} \right\|_2^2 \right) / d_b,
$$

$$
\tilde{v}_{t,b} = (1 - \beta_2) \left( \left\| \bar{g}_{t,(b)} \right\|_2^2 / d_b + \sigma_b^2 \right) + \beta_2 v_{t-1,b}.
$$

**Lemma 3.13** (first-order approximation). *With Assumption 3.10, it holds that for any block $b \in [B]$*

$$
\mathbb{E} \sum_{t=1}^{T} \frac{g_{t,(b)}^{\top} \bar{g}_{t,(b)}}{\sqrt{v_{t,b} + \epsilon}} \geq \frac{1}{2} \mathbb{E} \sum_{t=1}^{T} \frac{\left\| \bar{g}_{t,(b)} \right\|_2^2}{\sqrt{\tilde{v}_{t,b} + \epsilon}} - \sqrt{1 - \beta_2} T d_b \sigma_b - \frac{d_b \sigma_b \beta_2}{\sqrt{1 - \beta_2}} \mathbb{E} \left[ \ln \frac{v_{T,b} + \epsilon}{v_{0,b} + \epsilon} \right].
\tag{4}
$$

*Proof of Lemma 3.13.* The first order change in block $b$ can decomposed into two terms.

$$
\mathbb{E} \sum_{t=1}^{T} \sum_{\Phi(i)=b} \frac{g_{t,i} \bar{g}_{t,i}}{\sqrt{v_{t,b} + \epsilon}} = \mathbb{E} \sum_{t=1}^{T} \sum_{\Phi(i)=b} \frac{g_{t,i} \bar{g}_{t,i}}{\sqrt{\tilde{v}_{t,b} + \epsilon}} + \mathbb{E} \left[ \sum_{t=1}^{T} \sum_{\Phi(i)=b} \frac{g_{t,i} \bar{g}_{t,i}}{\sqrt{v_{t,b} + \epsilon}} - \frac{g_{t,i} \bar{g}_{t,i}}{\sqrt{\tilde{v}_{t,b} + \epsilon}} \right]
\tag{9}
$$

The first term $\mathbb{E}\sum_{t=1}^{T}\sum_{\Phi(i)=b}\frac{g_{t,i}\bar{g}_{t,i}}{\sqrt{\tilde{v}_{t,b}+\epsilon}}$ equals to $\mathbb{E}\sum_{t=1}^{T}\sum_{\Phi(i)=b}\frac{\bar{g}_{t,i}^2}{\sqrt{\tilde{v}_{t,b}+\epsilon}}$ when taking expectation conditional on $\boldsymbol{x}_{t-1}$. For the second term, it holds for each $t$ that

$$\sum_{\Phi(i)=b}\left|g_{t,i}\bar{g}_{t,i}\left(\frac{1}{\sqrt{v_{t,b}+\epsilon}}-\frac{1}{\sqrt{\tilde{v}_{t,b}+\epsilon}}\right)\right|$$

$$=\sum_{\Phi(i)=b}\frac{|g_{t,i}\bar{g}_{t,i}\left(\tilde{v}_{t,b}-v_{t,b}\right)|}{\sqrt{v_{t,b}+\epsilon}\sqrt{\tilde{v}_{t,b}+\epsilon}\left(\sqrt{v_{t,b}+\epsilon}+\sqrt{\tilde{v}_{t,b}+\epsilon}\right)}$$

$$=\sum_{\Phi(i)=b}\frac{\left|g_{t,i}\bar{g}_{t,i}(1-\beta_2)\left(\left\|\bar{\boldsymbol{g}}_{t,(b)}\right\|_2^2/d_b+\sigma_b^2-\left\|\boldsymbol{g}_{t,(b)}\right\|_2^2/d_b\right)\right|}{\sqrt{v_{t,b}+\epsilon}\sqrt{\tilde{v}_{t,b}+\epsilon}\left(\sqrt{v_{t,b}+\epsilon}+\sqrt{\tilde{v}_{t,b}+\epsilon}\right)}$$

$$=\sum_{\Phi(i)=b}\frac{\left|g_{t,i}\bar{g}_{t,i}(1-\beta_2)\left(\sqrt{\left\|\bar{\boldsymbol{g}}_{t,(b)}\right\|_2^2/d_b+\sigma_b^2}+\sqrt{\left\|\boldsymbol{g}_{t,(b)}\right\|_2^2/d_b}\right)\left(\sqrt{\left\|\bar{\boldsymbol{g}}_{t,(b)}\right\|_2^2/d_b+\sigma_b^2}-\sqrt{\left\|\boldsymbol{g}_{t,(b)}\right\|_2^2/d_b}\right)\right|}{\sqrt{v_{t,b}+\epsilon}\sqrt{\tilde{v}_{t,b}+\epsilon}\left(\sqrt{v_{t,b}+\epsilon}+\sqrt{\tilde{v}_{t,b}+\epsilon}\right)}$$

$$\overset{(1)}{\leq}\sum_{\Phi(i)=b}\frac{\left|g_{t,i}\bar{g}_{t,i}\sqrt{1-\beta_2}\left(\sqrt{\left\|\bar{\boldsymbol{g}}_{t,(b)}\right\|_2^2/d_b+\sigma_b^2}-\sqrt{\left\|\boldsymbol{g}_{t,(b)}\right\|_2^2/d_b}\right)\right|}{\sqrt{v_{t,b}+\epsilon}\sqrt{\tilde{v}_{t,b}+\epsilon}}$$

$$\overset{(2)}{\leq}\frac{1}{2}\sum_{\Phi(i)=b}\frac{\bar{g}_{t,i}^2}{\sqrt{\tilde{v}_{t,b}+\epsilon}}\frac{\left(\sqrt{\left\|\bar{\boldsymbol{g}}_{t,(b)}\right\|_2^2/d_b+\sigma_b^2}-\sqrt{\left\|\boldsymbol{g}_{t,(b)}\right\|_2^2/d_b}\right)^2}{\mathbb{E}\left[\left(\sqrt{\left\|\bar{\boldsymbol{g}}_{t,(b)}\right\|_2^2/d_b+\sigma_b^2}-\sqrt{\left\|\boldsymbol{g}_{t,(b)}\right\|_2^2/d_b}\right)^2|\boldsymbol{x}_{t-1}\right]} \tag{10}$$

$$+\frac{1}{2}\sum_{\Phi(i)=b}\frac{(1-\beta_2)g_{t,i}^2\mathbb{E}\left[\left(\sqrt{\left\|\bar{\boldsymbol{g}}_{t,(b)}\right\|_2^2/d_b+\sigma_b^2}-\sqrt{\left\|\boldsymbol{g}_{t,(b)}\right\|_2^2/d_b}\right)^2|\boldsymbol{x}_{t-1}\right]}{(v_{t,b}+\epsilon)\sqrt{\tilde{v}_{t,b}+\epsilon}} \tag{11}$$

The first inequality (1) is because $v_{t,b}+\epsilon\geq(1-\beta_2)\left\|\boldsymbol{g}_{t,(b)}\right\|_2^2/d_b$ and $\tilde{v}_{t,b}+\epsilon\geq(1-\beta_2)\left(\left\|\bar{\boldsymbol{g}}_{t,(b)}\right\|_2^2/d_b+\sigma_b^2\right)$. The second inequality (2) is obtained with AM-GM inequality. For the term in Equation 10, it will be exactly $\frac{1}{2}\frac{\left\|\bar{\boldsymbol{g}}_{t,(b)}\right\|_2^2}{\sqrt{\tilde{v}_{t,b}+\epsilon}}$ after taking expectation conditional on $\boldsymbol{x}_{t-1}$ because only $\left(\sqrt{\left\|\bar{\boldsymbol{g}}_{t,(b)}\right\|_2^2/d_b+\sigma_b^2}-\sqrt{\left\|\boldsymbol{g}_{t,(b)}\right\|_2^2/d_b}\right)^2$ depends on $\boldsymbol{x}_t$. For the term in Equation 11, we have the following inequality

$$\mathbb{E}\left[\left(\sqrt{\left\|\bar{\boldsymbol{g}}_{t,(b)}\right\|_2^2/d_b+\sigma_b^2}-\sqrt{\left\|\boldsymbol{g}_{t,(b)}\right\|_2^2/d_b}\right)^2\Bigg|\boldsymbol{x}_{t-1}\right]$$

$$=\mathbb{E}\left[\left\|\bar{\boldsymbol{g}}_{t,(b)}\right\|_2^2/d_b+\sigma_b^2+\sum_{\Phi(j)=b}g_{t,j}^2/d_b-2\sqrt{\left\|\boldsymbol{g}_{t,(b)}\right\|_2^2/d_b}\sqrt{\left\|\bar{\boldsymbol{g}}_{t,(b)}\right\|_2^2/d_b+\sigma_i^2}\Bigg|\boldsymbol{x}_{t-1}\right]$$

$$\overset{(1)}{\leq}2\left(\left\|\bar{\boldsymbol{g}}_{t,(b)}\right\|_2^2/d_b+\sigma_b^2\right)-2\sqrt{\left\|\bar{\boldsymbol{g}}_{t,(b)}\right\|_2^2/d_b+\sigma_b^2}\mathbb{E}\left[\sqrt{\left\|\boldsymbol{g}_{t,(b)}\right\|_2^2/d_b}\Bigg|\boldsymbol{x}_{t-1}\right]$$

$$\overset{(2)}{\leq}2\left(\left\|\bar{\boldsymbol{g}}_{t,(b)}\right\|_2^2/d_b+\sigma_b^2\right)-2\sqrt{\left\|\bar{\boldsymbol{g}}_{t,(b)}\right\|_2^2/d_b+\sigma_b^2}\sqrt{\left\|\bar{\boldsymbol{g}}_{t,(b)}\right\|_2^2/d_b}$$

$$=2\sqrt{\left\|\bar{\boldsymbol{g}}_{t,(b)}\right\|_2^2/d_b+\sigma_b^2}\left(\sqrt{\left\|\bar{\boldsymbol{g}}_{t,(b)}\right\|_2^2/d_b+\sigma_b^2}-\sqrt{\left\|\bar{\boldsymbol{g}}_{t,(b)}\right\|_2^2/d_b}\right)$$

$$\leq2\sqrt{\left\|\bar{\boldsymbol{g}}_{t,(b)}\right\|_2^2/d_b+\sigma_b^2}\sigma_b.$$

The first inequality (1) replaces $\mathbb{E}[\sum_{\Phi(j)=b}g_{t,j}^2/d_b\mid\boldsymbol{x}_{t-1}]$ with $\left\|\bar{\boldsymbol{g}}_{t,(b)}\right\|_2^2/d_b+\sigma_b^2$ based on Assumption 3.10. The second inequality (2) is because $\ell_2$ norm is a convex function. Then we can

bound Equation 11 by

$$\sum_{\Phi(i)=b} \frac{(1-\beta_2)g_{t,i}^2 \mathbb{E}\left[\left(\sqrt{\left\|\bar{\boldsymbol{g}}_{t,(b)}\right\|_2^2/d_b + \sigma_b^2} - \sqrt{\left\|\boldsymbol{g}_{t,(b)}\right\|_2^2/d_b}\right)^2 \mid \boldsymbol{x}_{t-1}\right]}{(v_{t,b}+\epsilon)\sqrt{\tilde{v}_{t,b}+\epsilon}}$$

$$\leq \sum_{\Phi(i)=b} \frac{(1-\beta_2)g_{t,i}^2 2\sqrt{\left\|\bar{\boldsymbol{g}}_{t,(b)}\right\|_2^2/d_b + \sigma_b^2}\,\sigma_b}{(v_{t,b}+\epsilon)\sqrt{\tilde{v}_{t,b}+\epsilon}}$$

$$\leq 2\sqrt{1-\beta_2}\sigma_b \sum_{\Phi(i)=b} \frac{g_{t,i}^2}{v_{t,b}+\epsilon}.$$

Then back to Equation 9, we have that

$$\mathbb{E}\sum_{t=1}^{T}\sum_{\Phi(i)=b} \frac{g_{t,i}\bar{g}_{t,i}}{\sqrt{v_{t,b}+\epsilon}} = \mathbb{E}\sum_{t=1}^{T}\sum_{\Phi(i)=b} \frac{g_{t,i}\bar{g}_{t,i}}{\sqrt{\tilde{v}_{t,b}+\epsilon}} + \mathbb{E}\left[\sum_{t=1}^{T}\sum_{\Phi(i)=b} \frac{g_{t,i}\bar{g}_{t,i}}{\sqrt{v_{t,b}+\epsilon}} - \frac{g_{t,i}\bar{g}_{t,i}}{\sqrt{\tilde{v}_{t,b}+\epsilon}}\right]$$

$$\geq \mathbb{E}\sum_{t=1}^{T}\sum_{\Phi(i)=b} \frac{\bar{g}_{t,i}^2}{\sqrt{\tilde{v}_{t,b}+\epsilon}} - \frac{1}{2}\mathbb{E}\sum_{t=1}^{T}\sum_{\Phi(i)=b} \frac{\bar{g}_{t,i}^2}{\sqrt{\tilde{v}_{t,b}+\epsilon}} - \frac{1}{2}2\sqrt{1-\beta_2}\sigma_b \mathbb{E}\sum_{t=1}^{T} \frac{\left\|\boldsymbol{g}_{t,(b)}\right\|_2^2}{v_{t,b}+\epsilon}$$

$$= \frac{1}{2}\mathbb{E}\sum_{t=1}^{T}\sum_{\Phi(i)=b} \frac{\bar{g}_{t,i}^2}{\sqrt{\tilde{v}_{t,b}+\epsilon}} - \sqrt{1-\beta_2}\sigma_b \mathbb{E}\sum_{t=1}^{T} \frac{\left\|\boldsymbol{g}_{t,(b)}\right\|_2^2}{v_{t,b}+\epsilon}.$$

For the second term, we can apply Lemma 3.12 and get that

$$\sum_{t=1}^{T} \frac{\left\|\boldsymbol{g}_{t,(b)}\right\|_2^2/d_b}{v_{t,b}+\epsilon} \leq T + \frac{\beta_2}{1-\beta_2}\ln\frac{v_{T,b}+\epsilon}{v_{0,b}+\epsilon}.$$

Combining these two terms, we can get that

$$\mathbb{E}\sum_{t=1}^{T}\sum_{\Phi(i)=b} \frac{g_{t,i}\bar{g}_{t,i}}{\sqrt{v_{t,b}+\epsilon}} \geq \frac{1}{2}\mathbb{E}\sum_{t=1}^{T}\sum_{\Phi(i)=b} \frac{\bar{g}_{t,i}^2}{\sqrt{\tilde{v}_{t,b}+\epsilon}} - \sqrt{1-\beta_2}Td_b\sigma_b - \frac{d_b\sigma_b\beta_2}{\sqrt{1-\beta_2}}\mathbb{E}\left[\ln\frac{v_{T,b}+\epsilon}{v_{0,b}+\epsilon}\right].$$

$$\square$$

Next we need Lemma 3.14 to deal with the denominator in the approximated first order term. The lemma is largely inspired by Lemma 6 in Li & Lin (2024), where we further generalize it to the case of block-wise `Adam`.

**Lemma 3.14.** *With Assumption 3.10, it holds that for any block $b \in [B]$*

$$\sum_{t=\frac{T}{2}+1}^{T} \mathbb{E}\left[\sqrt{\tilde{v}_{t,b}+\epsilon}\right] \leq \frac{2\beta_2^{\frac{T}{4}}}{1-\beta_2}\sqrt{v_{0,b}} + \frac{T}{2}\sigma_b + \frac{T}{2}\sqrt{\epsilon} + 2\sum_{t=1}^{T}\mathbb{E}\left[\frac{\left\|\bar{\boldsymbol{g}}_{t,(b)}\right\|_2^2/d_b}{\sqrt{\tilde{v}_{t,b}+\epsilon}}\right]. \tag{6}$$

*Proof of Lemma 3.14.* For each $t \leq T$, we have that

$$\mathbb{E}\left[\sqrt{\tilde{v}_{t,b}+\epsilon}\right]$$

$$= \mathbb{E}\left[\sqrt{\beta_2 v_{t-1,b} + (1-\beta_2)(\left\|\bar{\boldsymbol{g}}_{t,(b)}\right\|_2^2/d_b + \sigma_b^2) + \epsilon}\right]$$

$$= \mathbb{E}\left[\frac{\beta_2 v_{t-1,b} + (1-\beta_2)\sigma_b^2 + \epsilon}{\sqrt{\beta_2 v_{t-1,b} + (1-\beta_2)(\left\|\bar{\boldsymbol{g}}_{t,(b)}\right\|_2^2/d_b + \sigma_b^2) + \epsilon}}\right] + (1-\beta_2)\mathbb{E}\left[\frac{\left\|\bar{\boldsymbol{g}}_{t,(b)}\right\|_2^2/d_b}{\sqrt{\tilde{v}_{t,b}+\epsilon}}\right] \tag{12}$$

$$\leq \mathbb{E}\left[\sqrt{\beta_2 v_{t-1,b} + (1-\beta_2)\sigma_b^2 + \epsilon}\right] + (1-\beta_2)\mathbb{E}\left[\frac{\left\|\bar{\boldsymbol{g}}_{t,(b)}\right\|_2^2/d_b}{\sqrt{\tilde{v}_{t,b}+\epsilon}}\right].$$

We use $\beta_2 v_{t-1,b} + (1-\beta_2)(\sum_{\Phi(i)=b} \bar{g}_{t,i}^2/d_b + \sigma_b^2) + \epsilon \geq \beta_2 v_{t-1,b} + (1-\beta_2)\sigma_b^2 + \epsilon$ in the last step. And for each $s \leq t-1$, we have that

$$\mathbb{E}\left[\sqrt{\beta_2^s v_{t-s,b} + (1-\beta_2^s)\sigma_b^2 + \epsilon}\right]$$

$$=\mathbb{E}\left[\sqrt{\beta_2^{s+1} v_{t-s-1,b} + \beta_2^s(1-\beta_2)\left\|\boldsymbol{g}_{t-s,(b)}\right\|_2^2/d_b + (1-\beta_2^s)\sigma_b^2 + \epsilon}\right]$$

$$=\mathbb{E}\left[\mathbb{E}\left[\sqrt{\beta_2^{s+1} v_{t-s-1,b} + \beta_2^s(1-\beta_2)\left\|\boldsymbol{g}_{t-s,(b)}\right\|_2^2/d_b + (1-\beta_2^s)\sigma_b^2 + \epsilon}\Big|\boldsymbol{x}_{t-s-1}\right]\right]$$

$$\overset{(1)}{\leq}\mathbb{E}\left[\sqrt{\beta_2^{s+1} v_{t-s-1,b} + \beta_2^s(1-\beta_2)\mathbb{E}\left[\left\|\boldsymbol{g}_{t-s,(b)}\right\|_2^2/d_b\Big|\boldsymbol{x}_{t-s-1}\right] + (1-\beta_2^s)\sigma_b^2 + \epsilon}\right]$$

$$\overset{(2)}{\leq}\mathbb{E}\left[\sqrt{\beta_2^{s+1} v_{t-s-1,b} + \beta_2^s(1-\beta_2)\left\|\bar{\boldsymbol{g}}_{t-s,(b)}\right\|_2^2/d_b + (1-\beta_2^{s+1})\sigma_b^2 + \epsilon}\right]$$

$$=\mathbb{E}\left[\frac{\beta_2^{s+1} v_{t-s-1,b} + (1-\beta_2^{s+1})\sigma_b^2 + \epsilon}{\sqrt{\beta_2^{s+1} v_{t-s-1,b} + \beta_2^s(1-\beta_2)\left\|\bar{\boldsymbol{g}}_{t-s,(b)}\right\|_2^2/d_b + (1-\beta_2^{s+1})\sigma_b^2 + \epsilon}}\right]$$

$$+\mathbb{E}\left[\frac{\beta_2^s(1-\beta_2)\left\|\bar{\boldsymbol{g}}_{t-s,(b)}\right\|_2^2/d_b}{\sqrt{\beta_2^{s+1} v_{t-s-1,b} + \beta_2^s(1-\beta_2)\left\|\bar{\boldsymbol{g}}_{t-s,(b)}\right\|_2^2/d_b + (1-\beta_2^{s+1})\sigma_b^2 + \epsilon}}\right]$$

$$\leq\mathbb{E}\left[\sqrt{\beta_2^{s+1} v_{t-s-1,b} + (1-\beta_2^{s+1})\sigma_b^2 + \epsilon}\right] + \sqrt{\beta_2^s}(1-\beta_2)\mathbb{E}\left[\frac{\left\|\bar{\boldsymbol{g}}_{t-s,(b)}\right\|_2^2/d_b}{\sqrt{\tilde{v}_{t-s,b} + \epsilon}}\right].$$

The first inequality (1) is because $\sqrt{x}$ is a concave function. The second inequality (2) is based on the noise Assumption 3.10. By summing the above inequality over $s = 1, \cdots, t-1$, we have that

$$\mathbb{E}\left[\sqrt{\beta_2 v_{t-1,b} + (1-\beta_2)\sigma_b^2 + \epsilon}\right]$$

$$\leq\mathbb{E}\left[\sqrt{\beta_2^t v_{0,b} + (1-\beta_2^t)\sigma_b^2 + \epsilon}\right] + \sum_{s=1}^{t-1}\sqrt{\beta_2^s}(1-\beta_2)\mathbb{E}\left[\frac{\left\|\bar{\boldsymbol{g}}_{t-s,(b)}\right\|_2^2/d_b}{\sqrt{\tilde{v}_{t-s,b} + \epsilon}}\right]$$

$$\leq\sqrt{\beta_2^t v_{0,b}} + \sqrt{\sigma_b^2 + \epsilon} + \sum_{s=1}^{t-1}\sqrt{\beta_2^s}(1-\beta_2)\mathbb{E}\left[\frac{\left\|\bar{\boldsymbol{g}}_{t-s,(b)}\right\|_2^2/d_b}{\sqrt{\tilde{v}_{t-s,b} + \epsilon}}\right].$$

Back to Equation 12, we have that

$$\mathbb{E}\left[\sqrt{\tilde{v}_{t,b} + \epsilon}\right] \leq \sqrt{\beta_2^t v_{0,b}} + \sqrt{\sigma_b^2 + \epsilon} + \sum_{s=0}^{t-1}\sqrt{\beta_2^s}(1-\beta_2)\mathbb{E}\left[\frac{\left\|\bar{\boldsymbol{g}}_{t-s,(b)}\right\|_2^2/d_b}{\sqrt{\tilde{v}_{t-s,b} + \epsilon}}\right].$$

By summing the above inequality over $t = \frac{T}{2}+1, \cdots, T$, we have that

$$\sum_{t=\frac{T}{2}+1}^{T}\left[\sqrt{\tilde{v}_{t,b} + \epsilon}\right] \leq \sum_{t=\frac{T}{2}+1}^{T}\sqrt{\beta_2^t v_{0,b}} + \frac{T}{2}\sqrt{\sigma_b^2 + \epsilon} + \sum_{t=\frac{T}{2}+1}^{T}\sum_{s=0}^{t-1}\sqrt{\beta_2^s}(1-\beta_2)\mathbb{E}\left[\frac{\left\|\bar{\boldsymbol{g}}_{t-s,(b)}\right\|_2^2/d_b}{\sqrt{\tilde{v}_{t-s,b} + \epsilon}}\right]$$

$$\leq \frac{\beta_2^{\frac{T}{4}}}{1-\sqrt{\beta_2}}\sqrt{v_{0,b}} + \frac{T}{2}\sqrt{\sigma_b^2 + \epsilon} + \frac{1-\beta_2}{1-\sqrt{\beta_2}}\sum_{t=1}^{T}\mathbb{E}\left[\frac{\left\|\bar{\boldsymbol{g}}_{t,(b)}\right\|_2^2/d_b}{\sqrt{\tilde{v}_{t,b} + \epsilon}}\right]$$

$$= \frac{\beta_2^{\frac{T}{4}}(1+\sqrt{\beta_2})}{1-\beta_2}\sqrt{v_{0,b}} + \frac{T}{2}\sqrt{\sigma_b^2 + \epsilon} + (1+\sqrt{\beta_2})\sum_{t=1}^{T}\mathbb{E}\left[\frac{\left\|\bar{\boldsymbol{g}}_{t,(b)}\right\|_2^2/d_b}{\sqrt{\tilde{v}_{t,b} + \epsilon}}\right]$$

$$\leq \frac{2\beta_2^{\frac{T}{4}}}{1-\beta_2}\sqrt{v_{0,b}} + \frac{T}{2}\sigma_b + \frac{T}{2}\sqrt{\epsilon} + 2\sum_{t=1}^{T}\mathbb{E}\left[\frac{\left\|\bar{\boldsymbol{g}}_{t,(b)}\right\|_2^2/d_b}{\sqrt{\tilde{v}_{t,b} + \epsilon}}\right].$$

$\square$

This following Lemma D.1 is to control the growth of $v_{T,b}$ so that the right hand side in Lemma 3.12 is indeed $\Theta\left(T + \frac{\log T}{1-\beta_2}\right)$ instead of $\Theta(\frac{T}{1-\beta_2})$ when all the constants are poly$(T)$.

**Lemma D.1.** *Suppose there exists $\boldsymbol{H}$ dominates $\left|\nabla^2 L(\boldsymbol{x})\right|$ and follows the partition $\Phi$ (Definition 3.9). We denote the diagonal element in block $b$ by $H_b$, i.e., $\boldsymbol{H}_{i,i} = H_{\Phi(i)}$. With Assumption 3.10, it holds that*

$$\ln \frac{\mathbb{E}\max_{b\in[B]} v_{T,b} + \epsilon}{v_0 + \epsilon} \le 2\ln\left(1 + \frac{\sum_{b=1}^B \sigma_b^2 + \|\nabla L(\boldsymbol{x}_0)\|_\Phi^2 + \sum_{b\in[B]} H_b^2 d_b \eta^2 T(T + \frac{1}{1-\beta_2})}{v_0 + \epsilon}\right) + \ln 32$$

*Proof of Lemma D.1.* From the definition of $v_{t,b}$ and Assumption 3.10, we have that

$$\mathbb{E}\max_{b\in[B]} v_{t,b} = \mathbb{E}\max_{b\in[B]}\left[\beta_2^t v_{0,b} + (1-\beta_2)\sum_{s=1}^t \beta_2^{t-s}\left\|\boldsymbol{g}_{s,(b)}\right\|_2^2 / d_b\right]$$

$$\le \|\boldsymbol{v}_0\|_\infty + (1-\beta_2)\mathbb{E}\sum_{s=1}^t \beta_2^{t-s}\max_{b\in[B]}\left\|\boldsymbol{g}_{s,(b)}\right\|_2^2 / d_b \tag{13}$$

We can bound each $\left\|\boldsymbol{g}_{s,(b)}\right\|_2^2 / d_b$ as following

$$\left\|\boldsymbol{g}_{s,(b)}\right\|_2^2 / d_b \overset{(1)}{\le} 2/d_b\left\|\mathbb{E}[\boldsymbol{g}_{s,(b)}|\boldsymbol{x}_{s-1}]\right\|_2^2 + 2/d_b\left\|\boldsymbol{g}_{s,(b)} - \mathbb{E}[\boldsymbol{g}_{s,(b)}|\boldsymbol{x}_{s-1}]\right\|_2^2$$

$$= 2/d_b\left\|\nabla_{(b)} L(\boldsymbol{x}_{s-1})\right\|_2^2 + 2/d_b\left\|\boldsymbol{g}_{s,(b)} - \mathbb{E}[\boldsymbol{g}_{s,(b)}|\boldsymbol{x}_{s-1}]\right\|_2^2$$

$$\overset{(2)}{\le} 4/d_b\left\|\nabla_{(b)} L(\boldsymbol{x}_0)\right\|_2^2 + 4/d_b\left\|\nabla_{(b)} L(\boldsymbol{x}_{s-1}) - \nabla_{(b)} L(\boldsymbol{x}_0)\right\|_2^2 + 2/d_b\left\|\boldsymbol{g}_{s,(b)} - \mathbb{E}[\boldsymbol{g}_{s,(b)}|\boldsymbol{x}_{s-1}]\right\|_2^2$$

$$\le 4\max_{b'\in[B]}\left\|\nabla_{(b')} L(\boldsymbol{x}_0)\right\|_2^2 / d_{b'} + 4\left\|\nabla L(\boldsymbol{x}_{s-1}) - \nabla L(\boldsymbol{x}_0)\right\|_2^2 + 2\sum_{b'=1}^B\left\|\boldsymbol{g}_{s,(b')} - \mathbb{E}[\boldsymbol{g}_{s,(b')}|\boldsymbol{x}_{s-1}]\right\|_2^2 / d_{b'}$$

where we employ $(a+b)^2 \le 2a^2 + 2b^2$ in (1) and (2). This bound holds for any specific $b \in [B]$ so it is also an upper bound for $\max_{b\in[B]}\left\|\boldsymbol{g}_{s,(b)}\right\|_2^2 / d_b$. We will further simplify the bound and first upper bound the distance between gradients with the distance between parameters

$$\|\nabla L(\boldsymbol{x}_{s-1}) - \nabla L(\boldsymbol{x}_0)\|_2^2 \le \sum_{b=1}^B H_b\left\|\boldsymbol{x}_{s-1,(b)} - \boldsymbol{x}_{0,(b)}\right\|_2^2. \tag{14}$$

The distance between parameters can be characterized by the total updates and use Lemma 3.12 in (1) below

$$\frac{1}{d_b}\left\|\boldsymbol{x}_{t,(b)} - \boldsymbol{x}_{0,(b)}\right\|_2^2 = \frac{\eta^2}{d_b}\sum_{\Phi(j)=b}\left|\sum_{s=1}^t \frac{g_{s,j}}{\sqrt{v_{s,b}+\epsilon}}\right|^2 \le \frac{\eta^2}{d_b}\sum_{\Phi(j)=b} t\sum_{s=1}^t \frac{g_{s,j}^2}{v_{s,b}+\epsilon}$$

$$= \eta^2 t\sum_{s=1}^t \frac{\sum_{\Phi(j)=b} g_{s,j}^2/d_b}{v_{s,b}+\epsilon} \overset{(1)}{\le} \eta^2 t\left(t + \frac{\beta_2}{1-\beta_2}\ln\frac{v_{t,b}+\epsilon}{v_{0,b}+\epsilon}\right)$$

$$\le \eta^2 t^2 + \eta^2 t\frac{\beta_2}{1-\beta_2}\ln\frac{\max_{b'\in[B]} v_{t,b'}+\epsilon}{v_0+\epsilon}. \tag{15}$$

We can bound the distance between stochastic gradient and deterministic gradient with Assumption 3.10

$$\mathbb{E}\sum_{b'=1}^B\left\|\boldsymbol{g}_{s,(b')} - \mathbb{E}[\boldsymbol{g}_{s,(b')}|\boldsymbol{x}_{s-1}]\right\|_2^2 / d_{b'} \le \sum_{b=1}^B \sigma_b^2.$$

Combining these results, we can get that

$$(1-\beta_2)\mathbb{E}\sum_{s=1}^{t}\beta_2^{t-s}\max_{b\in[B]}\left\|\boldsymbol{g}_{s,(b)}\right\|_2^2/d_b$$

$$\leq(1-\beta_2)\sum_{s=1}^{t}\beta_2^{t-s}\left[4\max_{b'\in[B]}\left\|\nabla_{(b')}L(\boldsymbol{x}_0)\right\|_2^2/d_{b'}+4\sum_{b=1}^{B}H_bd_b^2\eta^2(s-1)^2+2\sum_{b=1}^{B}\sigma_b^2\right]$$

$$+(1-\beta_2)\sum_{s=1}^{t}\beta_2^{t-s}4\sum_{b=1}^{B}H_bd_b\eta^2(s-1)\frac{\beta_2}{1-\beta_2}\mathbb{E}\ln\frac{\max_{b'\in[B]}v_{t,b'}+\epsilon}{v_0+\epsilon}$$

$$\leq4\max_{b'\in[B]}\left\|\nabla_{(b')}L(\boldsymbol{x}_0)\right\|_2^2/d_{b'}+2\sum_{b=1}^{B}\sigma_b^2+4\sum_{b=1}^{B}H_bd_b^2\eta^2t^2+4\sum_{b=1}^{B}H_bd_b\eta^2t\frac{\beta_2}{1-\beta_2}\mathbb{E}\ln\frac{\max_{b'\in[B]}v_{t,b'}+\epsilon}{v_0+\epsilon}$$

We define $C=\epsilon+\left\|\boldsymbol{v}_0\right\|_\infty+2\sum_{b=1}^{B}\sigma_b^2+4\max_{b\in[B]}\left\|\nabla_{(b)}L(\boldsymbol{x}_0)\right\|_2^2/d_b$ for simplicity. We also define $G=\max_{1\leq t\leq T}\mathbb{E}\max_{b\in[B]}v_{t,b}+\epsilon$. There exists $t\leq T$ such that

$$G=\mathbb{E}\max_{b\in[B]}v_{t,b}+\epsilon\leq C+4\eta^2T^2\sum_{b=1}^{B}H_b^2d_b+4\eta^2T\frac{\beta_2}{1-\beta_2}\sum_{b=1}^{B}H_b^2d_b\ln\frac{G}{v_0+\epsilon}.$$

We will upper bound the last term by the linear term $G$ as following

$$4\eta^2T^2\sum_{b=1}^{B}H_b^2d_b+4\eta^2T\frac{\beta_2}{1-\beta_2}\sum_{b=1}^{B}H_b^2d_b\ln\frac{G}{v_0+\epsilon}$$

$$=4\eta^2T\frac{\beta_2}{1-\beta_2}\sum_{b\in[B]}H_b^2d_b\left(\ln\frac{G(1-\beta_2)}{4\sum_{b\in[B]}H_b^2d_b\eta^2T\beta_2}+\ln 4\sum_{b\in[B]}H_b^2d_b\eta^2T\frac{\beta_2}{(1-\beta_2)(v_0+\epsilon)}\right)$$

$$\leq\frac{G}{2}+4\eta^2T\frac{\beta_2}{1-\beta_2}\sum_{b\in[B]}H_b^2d_b\ln 4\sum_{b\in[B]}H_b^2d_b\eta^2T\frac{\beta_2}{(1-\beta_2)(v_0+\epsilon)}$$

$$\leq\frac{G}{2}+(v_0+\epsilon)\left(4\sum_{b\in[B]}H_b^2d_b\eta^2T\frac{\beta_2}{(1-\beta_2)(v_0+\epsilon)}\right)^2.$$

The last two inequalities come from $\ln x\leq\frac{x}{2}$ and $x\ln(x)\leq x^2$. Then we can get that

$$G\leq 2C+8\sum_{b\in[B]}H_b^2d_b\eta^2T^2+32(v_0+\epsilon)\left(\sum_{b\in[B]}H_b^2d_b\eta^2T\frac{\beta_2}{(1-\beta_2)(v_0+\epsilon)}\right)^2$$

and

$$\ln\frac{\mathbb{E}\max_{b\in[B]}v_{T,b}+\epsilon}{v_0+\epsilon}$$

$$\leq\ln\frac{2C+8\sum_{b\in[B]}H_b^2d_b\eta^2T^2+32(v_0+\epsilon)\left(\sum_{b\in[B]}H_b^2d_b\eta^2T\frac{\beta_2}{(1-\beta_2)(v_0+\epsilon)}\right)^2}{v_0+\epsilon}$$

$$\leq\ln\left[2\left(1+\frac{\sum_{b=1}^{B}\sigma_b^2+2\left\|\nabla L(\boldsymbol{x}_0)\right\|_\Phi^2}{v_0+\epsilon}+\frac{2\sum_{b\in[B]}H_b^2d_b\eta^2T^2}{v_0+\epsilon}+\frac{4\sum_{b\in[B]}H_b^2d_b\eta^2T\beta_2}{(1-\beta_2)(v_0+\epsilon)}\right)^2\right]$$

$$\leq2\ln\left(1+\frac{\sum_{b=1}^{B}\sigma_b^2+\left\|\nabla L(\boldsymbol{x}_0)\right\|_\Phi^2+\sum_{b\in[B]}H_b^2d_b\eta^2T(T+\frac{1}{1-\beta_2})}{v_0+\epsilon}\right)+\ln 32.$$

$\square$

Finally, we give the proof for Theorem 3.11. When $\Phi(i) = i$, i.e., each parameter forms a single block, it becomes the proof for Theorem 3.5.

**Theorem 3.11** (Main, Blockwise `Adam`). *For a specific partition $\Phi$, we consider the updates defined in Algorithm 3. Under Assumption 3.10, we have that*

$$\min_{\frac{T}{2} < t \leq T} \mathbb{E} \sum_{b=1}^{B} \sqrt{d_b} \left\| \nabla_{(b)} L(\boldsymbol{x}_t) \right\|_2 \leq 2\sqrt{2}E + \sqrt{2E} \sqrt{\frac{4\beta_2^{\frac{T}{4}}}{T(1-\beta_2)} d\sqrt{v_0} + \sum_{b=1}^{B} d_b \sigma_b + d\sqrt{\epsilon}}$$

*with $E = \frac{2}{\eta T} \mathbb{E}\left[L(\boldsymbol{x}_0) - L(\boldsymbol{x}_T)\right] + \left(1 + \frac{\beta_2 F}{T(1-\beta_2)}\right)\left(\eta H(L, \Phi) + 2\sqrt{1-\beta_2}\sum_{b=1}^{B} d_b \sigma_b\right)$ and $F = 2\ln\left(1 + \frac{\sum_{b=1}^{B}\sigma_b^2 + \|\nabla L(\boldsymbol{x}_0)\|_\Phi^2 + \sum_{b\in[B]} H_b^2 d_b \eta^2 T(T+\frac{1}{1-\beta_2})}{v_0+\epsilon}\right) + \ln 32.$*

*Proof of Theorem 3.11.* From Definition 3.9, there exists a diagonal matrix $\boldsymbol{H}$ that follows $\Phi$ and always dominates $\nabla^2 L(\boldsymbol{x})$ satisfying $H(L, \Phi) = \text{Tr}(\boldsymbol{H}) = \sum_{b\in[B]} H_b d_b$. In a single step, we can have that

$$L(\boldsymbol{x}_t) - L(\boldsymbol{x}_{t-1}) \leq \nabla L(\boldsymbol{x}_{t-1})^\top (\boldsymbol{x}_t - \boldsymbol{x}_{t-1}) + \frac{1}{2}\sum_{b=1}^{B} H_b \sum_{\Phi(i)=b} (x_{t,i} - x_{t-1,i})^2$$

$$= -\eta \sum_{b=1}^{B} \frac{\bar{\boldsymbol{g}}_{t,(b)}^\top \boldsymbol{g}_{t,(b)}}{\sqrt{v_{t,b}+\epsilon}} + \frac{1}{2}\eta^2 \sum_{b=1}^{B} H_b \frac{\left\|\boldsymbol{g}_{t,(b)}\right\|_2^2}{v_{t,b}+\epsilon}.$$

If we sum over $t$ from 1 to $T$ and take expectation, we can get

$$\mathbb{E}\left[L(\boldsymbol{x}_T) - L(\boldsymbol{x}_0)\right] \leq -\mathbb{E}\left[\eta \sum_{t=1}^{T}\sum_{b=1}^{B} \frac{\bar{\boldsymbol{g}}_{t,(b)}^\top \boldsymbol{g}_{t,(b)}}{\sqrt{v_{t,b}+\epsilon}}\right] + \frac{1}{2}\eta^2 \mathbb{E}\left[\sum_{t=1}^{T}\sum_{b=1}^{B} H_b \frac{\left\|\boldsymbol{g}_{t,(b)}\right\|_2^2}{v_{t,b}+\epsilon}\right]$$

$$\leq -\mathbb{E}\left[\eta \sum_{t=1}^{T}\sum_{b=1}^{B} \frac{\bar{\boldsymbol{g}}_{t,(b)}^\top \boldsymbol{g}_{t,(b)}}{\sqrt{v_{t,b}+\epsilon}}\right] + \frac{1}{2}\eta^2 \mathbb{E}\left[\sum_{b=1}^{B} H_b d_b \left(T + \frac{\beta_2}{1-\beta_2}\ln\frac{v_{T,b}+\epsilon}{v_{0,b}+\epsilon}\right)\right].$$

The second inequality comes from applying Lemma 3.12. By Lemma 3.13, we have that

$$\frac{1}{T}\mathbb{E}\left[\sum_{t=1}^{T} \frac{\left\|\bar{\boldsymbol{g}}_{t,(b)}\right\|_2^2}{\sqrt{\tilde{v}_{t,b}+\epsilon}}\right] \leq \frac{2}{\eta T}\mathbb{E}\left[L(\boldsymbol{x}_0) - L(\boldsymbol{x}_T)\right] + \frac{\eta}{T}\mathbb{E}\left[\sum_{b=1}^{B} H_b d_b\left(T + \frac{\beta_2}{1-\beta_2}\ln\frac{v_{T,b}+\epsilon}{v_{0,b}+\epsilon}\right)\right]$$

$$+ \frac{2}{T}\sum_{b=1}^{B} d_b \sigma_b \sqrt{1-\beta_2}\left(T + \frac{\beta_2}{1-\beta_2}\mathbb{E}\ln\frac{v_{T,b}+\epsilon}{v_{0,b}+\epsilon}\right)$$

$$\leq \frac{2}{\eta T}\mathbb{E}\left[L(\boldsymbol{x}_0) - L(\boldsymbol{x}_T)\right] + \eta\sum_{b=1}^{B} H_b d_b + 2\sqrt{1-\beta_2}\sum_{b=1}^{B} d_b\sigma_b$$

$$+ \frac{\beta_2}{T(1-\beta_2)}\left(\eta\sum_{b=1}^{B} H_b d_b + 2\sqrt{1-\beta_2}\sum_{b=1}^{B}\sigma_b\right)\max_{b\in[B]}\mathbb{E}\ln\frac{v_{T,b}+\epsilon}{v_{0,b}+\epsilon}$$

$$\leq \frac{2}{\eta T}\mathbb{E}\left[L(\boldsymbol{x}_0) - L(\boldsymbol{x}_T)\right] + \eta\sum_{b=1}^{B} H_b d_b + 2\sqrt{1-\beta_2}\sum_{b=1}^{B} d_b\sigma_b$$

$$+ \frac{\beta_2}{T(1-\beta_2)}\left(\eta\sum_{b=1}^{B} H_b d_b + 2\sqrt{1-\beta_2}\sum_{b=1}^{B} d_b\sigma_b\right)\ln\frac{\mathbb{E}\max_{b\in[B]} v_{T,b}+\epsilon}{v_0+\epsilon}.$$

From Lemma D.1, we can define

$$E = \frac{2}{\eta T}\mathbb{E}\left[L(\boldsymbol{x}_0) - L(\boldsymbol{x}_T)\right] + \left(1 + \frac{\beta_2 F}{T(1-\beta_2)}\right)\left(\eta\sum_{b=1}^{B} H_b d_b + 2\sqrt{1-\beta_2}\sum_{b=1}^{B} d_b\sigma_b\right),$$

with

$$F = 2 \ln \left( 1 + \frac{\sum_{b=1}^{B} \sigma_b^2 + \|\nabla L(\boldsymbol{x}_0)\|_\Phi^2 + \sum_{b \in [B]} H_b^2 d_b \eta^2 T(T + \frac{1}{1-\beta_2})}{v_0 + \epsilon} \right) + \ln 32.$$

Then it holds that

$$\frac{1}{T} \mathbb{E} \left[ \sum_{t=1}^{T} \sum_{b=1}^{B} \frac{\bar{\boldsymbol{g}}_{t,(b)}^2}{\sqrt{\tilde{v}_{t,b} + \epsilon}} \right] \le E.$$

By Lemma 3.14 and Cauchy inequality, we have that

$$\frac{2}{T} \mathbb{E} \sum_{t=\frac{T}{2}+1}^{T} \sum_{b=1}^{B} \sqrt{d_b} \left\| \bar{\boldsymbol{g}}_{t,(b)} \right\|_2 \le \left( \frac{2}{T} \mathbb{E} \sum_{t=\frac{T}{2}+1}^{T} \sum_{b=1}^{B} \frac{\left\| \bar{\boldsymbol{g}}_{t,(b)} \right\|_2^2}{\sqrt{\tilde{v}_{t,b} + \epsilon}} \right)^{\frac{1}{2}} \left( \frac{2}{T} \mathbb{E} \sum_{t=\frac{T}{2}+1}^{T} \sum_{b=1}^{B} d_b \sqrt{\tilde{v}_{t,b} + \epsilon} \right)^{\frac{1}{2}}$$

$$\le \sqrt{2E} \left( 4E + \frac{4\beta_2^{\frac{T}{4}}}{T(1-\beta_2)} d\sqrt{v_0} + \sum_{b=1}^{B} d_b \sigma_b + d\sqrt{\epsilon} \right)^{\frac{1}{2}}$$

$$\le 2\sqrt{2}E + \sqrt{2}\sqrt{E} \sqrt{\frac{4\beta_2^{\frac{T}{4}}}{T(1-\beta_2)} d\sqrt{v_0} + \sum_{b=1}^{B} d_b \sigma_b + d\sqrt{\epsilon}}.$$

This completes the proof. $\qquad\qquad\square$

## D.2 Proof for convergence rate of Adam

We rely on Definition D.2 that generalizes Definition 3.3 to obtain Theorem 3.5 from Theorem 3.11.

**Definition D.2** (Generalized version of Definition 3.3). *For any partition function $\Phi : [d] \to [B]$ and $\boldsymbol{H} = (H_1, \ldots, H_B)$, we say a function $L$ is $\boldsymbol{H}$-blockwisely-smooth w.r.t. $\Phi$-norm, if and only if $\left\| \nabla_{(b)} L(\boldsymbol{x}) - \nabla_{(b)} L(\boldsymbol{y}) \right\|_2 \le \sqrt{d_b} H_b \left\| \boldsymbol{x} - \boldsymbol{y} \right\|_\Phi$ for any $b \in [B]$, $\boldsymbol{x}, \boldsymbol{y} \in \mathbb{R}^d$.*

The following Lemma D.3 will show that $H(L, \Phi)$ is upper bounded by $\sum_{b=1}^{B} d_b H_b$ when $L$ is $\boldsymbol{H}$-blockwisely-smooth w.r.t. $\Phi$-norm.

**Lemma D.3.** *For any twice differentiable loss which is $\boldsymbol{H}$-blockwisely-smooth w.r.t. $\Phi$-norm (Definition D.2), we have for any $\boldsymbol{x}$ and $\boldsymbol{\Delta} \in \mathbb{R}^d$,*

$$\left| \boldsymbol{\Delta}^\top \nabla^2 L(\boldsymbol{x}) \boldsymbol{\Delta} \right| \le \sum_{b=1}^{B} H_b \left\| \boldsymbol{\Delta}_{(b)} \right\|_2^2. \tag{16}$$

*Then the diagonal matrix $\boldsymbol{A}$ defined by $\boldsymbol{A}_{i,i} = H_{\Phi(i)}$ follows partition $\Phi$ and dominates $\left| \nabla^2 L(\boldsymbol{x}) \right|$.*

*Proof of Lemma D.3.* From Definition D.2, we know that

$$H_b \geq \sup_{\boldsymbol{x},\boldsymbol{\Delta}} \frac{\left\| \nabla_{(b)} L(\boldsymbol{x} + \boldsymbol{\Delta}) - \nabla_{(b)} L(\boldsymbol{x}) \right\|_2}{\sqrt{d_b} \max_{b' \in [B]} \frac{\left\| \boldsymbol{\Delta}_{(b')} \right\|_2}{\sqrt{d_{b'}}}}$$

$$= \sup_{\boldsymbol{x},\boldsymbol{\Delta}} \frac{\left\| \nabla^2_{(b),:} L(\boldsymbol{x}) \boldsymbol{\Delta} \right\|_2}{\sqrt{d_b} \max_{b' \in [B]} \frac{\left\| \boldsymbol{\Delta}_{(b')} \right\|_2}{\sqrt{d_{b'}}}}$$

$$= \sup_{\boldsymbol{x},\boldsymbol{\Delta}} \frac{\left\| \sum_{b'=1}^B \nabla^2_{(b),(b')} L(\boldsymbol{x}) \boldsymbol{\Delta}_{(b')} \right\|_2}{\sqrt{d_b} \max_{b' \in [B]} \frac{\left\| \boldsymbol{\Delta}_{(b')} \right\|_2}{\sqrt{d_{b'}}}}$$

$$= \sup_{\boldsymbol{x},\boldsymbol{\Delta}, \left\| \boldsymbol{\Delta}'_{(b)} \right\|_2 \leq 1} \frac{\left\langle \boldsymbol{\Delta}'_{(b)}, \sum_{b'=1}^B \nabla^2_{(b),(b')} L(\boldsymbol{x}) \boldsymbol{\Delta}_{(b')} \right\rangle}{\sqrt{d_b} \max_{b' \in [B]} \frac{\left\| \boldsymbol{\Delta}_{(b')} \right\|_2}{\sqrt{d_{b'}}}}$$

$$= \sup_{\boldsymbol{x}, \left\| \boldsymbol{\Delta}_{(b')} \right\|_2 \leq \sqrt{d_{b'}}, \left\| \boldsymbol{\Delta}'_{(b)} \right\|_2 \leq 1} \frac{1}{\sqrt{d_b}} \left\langle \boldsymbol{\Delta}'_{(b)}, \sum_{b'=1}^B \nabla^2_{(b),(b')} L(\boldsymbol{x}) \boldsymbol{\Delta}_{(b')} \right\rangle$$

$$= \sup_{\boldsymbol{x},\boldsymbol{\Delta},\boldsymbol{\Delta}'} \sum_{b'=1}^B \frac{\sqrt{d_{b'}}}{\sqrt{d_b} \left\| \boldsymbol{\Delta}'_{(b)} \right\|_2 \left\| \boldsymbol{\Delta}_{(b')} \right\|_2} \left\langle \boldsymbol{\Delta}'_{(b)}, \nabla^2_{(b),(b')} L(\boldsymbol{x}) \boldsymbol{\Delta}_{(b')} \right\rangle.$$

Then for any $\boldsymbol{x}$ and $\boldsymbol{\Delta}$, we know that

$$H_b \left\| \boldsymbol{\Delta}_{(b)} \right\|_2^2 \geq \left\| \boldsymbol{\Delta}_{(b)} \right\|_2^2 \sum_{b'=1}^B \frac{\sqrt{d_{b'}}}{\sqrt{d_b} \left\| \boldsymbol{\Delta}_{(b)} \right\|_2 \left\| \boldsymbol{\Delta}_{(b')} \right\|_2} \left| \left\langle \boldsymbol{\Delta}_{(b)}, \nabla^2_{(b),(b')} L(\boldsymbol{x}) \boldsymbol{\Delta}_{(b')} \right\rangle \right|$$

$$= \sum_{b'=1}^B \frac{\sqrt{d_{b'}} \left\| \boldsymbol{\Delta}_{(b)} \right\|_2}{\sqrt{d_b} \left\| \boldsymbol{\Delta}_{(b')} \right\|_2} \left| \left\langle \boldsymbol{\Delta}_{(b)}, \nabla^2_{(b),(b')} L(\boldsymbol{x}) \boldsymbol{\Delta}_{(b')} \right\rangle \right|$$

and

$$2 \sum_{b=1}^B H_b \left\| \boldsymbol{\Delta}_{(b)} \right\|_2^2$$

$$= \sum_{b=1}^B H_b \left\| \boldsymbol{\Delta}_{(b)} \right\|_2^2 + \sum_{b'=1}^B H_{b'} \left\| \boldsymbol{\Delta}_{(b')} \right\|_2^2$$

$$\geq \sum_{b=1}^B \sum_{b'=1}^B \frac{\sqrt{d_{b'}} \left\| \boldsymbol{\Delta}_{(b)} \right\|_2}{\sqrt{d_b} \left\| \boldsymbol{\Delta}_{(b')} \right\|_2} \left| \left\langle \boldsymbol{\Delta}_{(b)}, \nabla^2_{(b),(b')} L(\boldsymbol{x}) \boldsymbol{\Delta}_{(b')} \right\rangle \right| + \sum_{b'=1}^B \sum_{b=1}^B \frac{\sqrt{d_b} \left\| \boldsymbol{\Delta}_{(b')} \right\|_2}{\sqrt{d_{b'}} \left\| \boldsymbol{\Delta}_{(b)} \right\|_2} \left| \left\langle \boldsymbol{\Delta}_{(b')}, \nabla^2_{(b'),(b)} L(\boldsymbol{x}) \boldsymbol{\Delta}_{(b)} \right\rangle \right|$$

$$\geq 2 \sum_{b=1}^B \sum_{b'=1}^B \left| \boldsymbol{\Delta}_{(b)}^\top \nabla^2_{(b),(b')} L(\boldsymbol{x}) \boldsymbol{\Delta}_{(b')} \right| \geq 2 \left| \boldsymbol{\Delta}^\top \nabla^2 L(\boldsymbol{x}) \boldsymbol{\Delta} \right|.$$

$\square$

# E  EXPERIMENT DETAILS

## E.1  TRAINING DETAILS

In `Adam` and its variants (including AdaSGD) we set $(\beta_1, \beta_2) = (0.9, 0.99)$ for experiments on the quadratic loss (Section 4.1) and ResNet18 (Section 4.3) and $(\beta_1, \beta_2) = (0.9, 0.95)$ for experiments on GPT-2 (Section 4.2). Momentum in SGD is also set to 0.9. Weight decay is always deactivated.

For the quadratic loss experiments in Section 4.1, we generate orthogonal matrices $\mathcal{R}_i$ in the following way. We first sample $\mathcal{M} \in \mathbb{R}^{d \times d}$ where $\mathcal{M}_{i,j}$ is i.i.d. sampled from $N(0,1)$. Then

$\mathcal{A} = \mathcal{M} - \mathcal{M}^\top$ is a skew-symmetric matrix and $\exp(t\mathcal{A})$ represents a continuous family of matrices. We define $\mathcal{R}_i = \exp(t_i\mathcal{A})$ for different $t_i$. When $t_i = 0$, we know $\mathcal{R}_i = I$. When $t_i \to \infty$, $\mathcal{R}_i$ converges to a random orthogonal matrix in distribution. We pick $t_1 = 0.002, t_2 = 0.008, t_3 = 0.015, t_4 = 0.1$ for our experiments. The initial $\boldsymbol{x}_0$ is decided by sampling from $\text{Unif}([0,1]^{1000})$ when there is no rotation. When $\mathcal{R}_i$ is not identity matrix, we will start training from $\mathcal{R}_i^\top \boldsymbol{x}_0$ to ensure the initial loss values are the same across different rotations.

For the experiments in Section 4.2, we train GPT-2 small (124M parameters)[4] on the OpenWebText corpus containing more than 9B tokens for 100k iterations with sequence length of $512$ sequence length and $480$ sentences per batch. We use cosine learning rate schedule of the same peak learning rate $6 \times 10^{-4}$ for all the adaptive optimizers, which is also the default of `nanoGPT` codebase. We did a grid search to find the maximum possible peak learning rate for `SGD`[5].

For the experiments in Section 4.3, we applied random crop and random horizontal flip augmentations over the training data to promote better generalization. We tuned each optimizer through searching over the same grid of learning rates[6] The number of iterations is adjusted per batch size to result in 20 epochs for each training run (for instance, 4000 iterations were used for a batch size of 256, and 1000 iterations were used for a batch size of 1024). For the training loss and training accuracy plotted in Figure 2, it is measured on a subset of augmented training data that is the same size of evaluation set. The evaluation loss and accuracy are measured on the entire evaluation set without the augmentation. Track running stats is set to false at initialization.

## E.2 `Adam` ON A ROTATED LOSS

A key difficulty in implementing `rotated Adam` arises from applying an orthogonal rotation on the parameters before calculating the loss. It is computationally infeasible to apply a 125M × 125M orthogonal matrix on the 125M-sized parameter vector. To avoid such computation, we design a new orthogonal transformer to rotate the parameters of the network. In what follows, we elaborate on this rotation.

`RandPerm`. Given a vector $v$ of size $d$, we can orthogonally rotate it by repeatedly applying these consecutive operations: 1. Permute the entries of the vector according to a randomly chosen permutation $\pi \in \mathbb{S}_d$. 2. Reshape the permuted vector into a 3D tensor of size $[s_1, s_2, s_3]$, apply a fixed orthogonal rotation of size $s \times s$ on each side of the tensor and then reshape it back to a vector of size $d$.

This operation performs an orthogonal transformation $\mathcal{R}$ on the input vector $v$. We can chain multiple operations of this kind and construct `RandPerm`$^k$, where $k$ is a positive number indicating the number of consecutive `RandPerm`s applied. Building upon this rotation, we train GPT-2 125M with `Adam` on $L \circ \text{RandPerm}^2$ to analyze our hypothesis regarding the $\ell_\infty$ geometry of the loss landscape and to verify that `Adam` will indeed suffer from the induced orthogonal equivariance. Figure 1 confirms our findings, as the performance of `rotated Adam` with `RandPerm`$^2$ is significantly worse than `Adam`. This suggests that `Adam` is highly sensitive to the rotation and adaptivity alone can't explain its advantage.

Concurrent work (Maes et al., 2024) also run `Adam` on a rotated loss. They use a different way to achieve global rotation efficiently. They also conduct extensive module-wise rotation experiments for a more fine-grained analysis.

## E.3 COMPUTATION OF MATRIX NORMS

As mentioned in Section 3.3, it is computationally infeasible to get the full Hessian matrix and directly compute norms of it. Instead we leverage Hessian vector product function in Jax to probe the Hessian matrix. We use Lanczos algorithm to estimate spectral norm.

---

[4]Our codebase is built upon `nanoGPT` codebase https://github.com/karpathy/nanoGPT.

[5]We tried $0.00001, 0.00003, 0.0001, 0.0003, 0.001, 0.003, 0.01, 0.03, 0.1, 0.3, 1$.

[6]We used the following values: $6.25 \times 10^{-4}, 1.25 \times 10^{-3}, 2.5 \times 10^{-3}, 5.0 \times 10^{-3}, 1.0 \times 10^{-2}, 2.0 \times 10^{-2}, 4.0 \times 10^{-2}, 8.0 \times 10^{-2}, 1.6 \times 10^{-1}, 3.2 \times 10^{-1}, 6.4 \times 10^{-1}, 1.28 \times 10^{0}$.

We propose Algorithm 4 to estimate the sum of absolute values for each row and sum over all the rows to get $(1, 1)$-norm of Hessian matrix. We first subsample a fixed batch of training data for estimating the $(1, 1)$-norm of Hessian matrix. The high-level idea is to compute the matrix vector products between Hessian of training loss on this batch and a sequence of random Cauchy vectors. Then we take the $\ell_1$ norm of the coordinate-wise median of the resulting sequence of Hessian vector products. Because the Cauchy distribution is 1-stable, the resulting product is also a vector of Cauchy random variables, and the magnitude of each element equals to $\ell_1$ norm of the corresponding row of the Hessian. Thus with infinitely many samples, the $\ell_1$ norm of the coordinate-wise median converges almost surely to the $(1, 1)$-norm of the Hessian.

We choose $n = 200$ for the measurement experiments on GPT-2 and $n = 50$ for the measurement experiments on ResNet18. We also prove a non-asymptotic high-probability multiplicative bound for the estimation error which depends mildly on the dimension $d$ in Theorem E.2.

---

**Algorithm 4** Estimation of $(1, 1)$-Norm of Hessian, $\nabla^2 L(\boldsymbol{x})$

---

**Input:** Number of Cauchy vectors $n$, parameter $\boldsymbol{x} \in \mathbb{R}^d$, loss $L$
1: **for** $i = 1$ to $n$ :
2:      Sample a independent Cauchy vector $v^{(i)} \in \mathbb{R}^d$ where $v_j^{(i)} \overset{\text{i.i.d.}}{\sim}$ Cauchy$(0, 1)$ for $j = 1, \ldots, d$.
3:      $\mathbf{H}_{:,i} \leftarrow \nabla^2 L(\boldsymbol{x}) \cdot v^{(i)}$                      (Using hessian-vector product)
4: **return** $\sum_{j=1}^d \text{median}(|\mathbf{H}_{j,:}|)$

---

First we prove that median of random variables following uniform distribution is sub-Gaussian.

**Lemma E.1.** *Suppose $Z_1, \cdots, Z_n \overset{\text{iid}}{\sim}$ Unif$([0, 1])$. Then*

$$P\left(\left|\text{median}(Z_1, \cdots, Z_n) - \frac{1}{2}\right| \geq \epsilon\right) \leq 2 \exp\left(-2n\epsilon^2\right)$$

*for any $\epsilon \geq 0$.*

*Proof of Lemma E.1.* Define $S = \sum_{i=1}^n \mathbf{1}_{Z_i \leq \frac{1}{2} - \epsilon}$. Since $\mathbf{1}_{Z_i \leq \frac{1}{2} - \epsilon}$ follows i.i.d. Bernoulli distribution with $p_1 = \frac{1}{2} - \epsilon$, $S \sim \text{Bin}(n, p_1)$.

$M_n = \text{median}(Z_1, \cdots, Z_n) \leq \frac{1}{2} - \epsilon$ if and only if at least $\frac{n+1}{2}$ $Z_i$'s are smaller than $\frac{1}{2} - \epsilon$. And we can apply Hoeffding's inequality on $S$ and get that

$$P(M_n \leq \frac{1}{2} - \epsilon) \leq P(S \geq \frac{n}{2}) = P(S - np_1 \geq \frac{n}{2} - np_1)$$

$$\leq \exp\left(-\frac{2(\frac{n}{2} - np_1)^2}{n}\right) = \exp\left(-2n\epsilon^2\right).$$

We can know $P(M_n \geq \frac{1}{2} + \epsilon) \leq \exp(-2n\epsilon^2)$ from the symmetry of distribution. $\qquad \square$

**Theorem E.2.** *For the estimate in Algorithm 4 with $n$ Cauchy vectors, it holds that*

$$P\left(\left|\sum_{j=1}^d \text{median}(|\mathbf{H}_{j,:}|) - \left\|\nabla^2 L(\boldsymbol{x})\right\|_{1,1}\right| \geq \epsilon \left\|\nabla^2 L(\boldsymbol{x})\right\|_{1,1}\right)$$

$$\leq 2d \exp(-\frac{n\Delta^2}{2}) + 2 \exp\left(-\frac{2n\epsilon^2 \cos^4((1+\Delta)\frac{\pi}{4})}{\pi^2}\right)$$

*for every $\epsilon, \Delta \in (0, 1)$ when $n \geq \frac{\pi^3}{2\epsilon^2 \cos^4((1+\Delta)\frac{\pi}{4})}$.*

In other words, for any $\epsilon, \delta \in (0, 1)$, we can use $n = \Omega(\ln d + \frac{1}{\epsilon^2} \ln \frac{1}{\delta})$ hessian-vector product of the loss $L$ at parameter $\boldsymbol{x}$ and $nd$ extra computation time to get an estimation of $(1,1)$-norm of $L$ with at most $\epsilon$ multiplicative error and at least probability $1 - \delta$.

*Proof of Theorem E.2.* Define $a_j = \sum_{k=1}^{d} |\nabla^2 L(\boldsymbol{x})_{j,k}|$ for $j \in [d]$. When $v_j^{(i)} \overset{\text{i.i.d.}}{\sim} \text{Cauchy}(0,1)$, it holds that $H_{j,i} = \nabla^2 L(\boldsymbol{x})_{j,:} \cdot v^{(i)}$ follows $\text{Cauchy}(0, a_j)$ because Cauchy distribution is 1-stable. And $H_{j,1}, \cdots, H_{j,n}$ are independent. Then it suffices to show that

$$P\left( \left| \sum_{j=1}^{d} a_j \text{median}(|Y_{j,1}|, \cdots, |Y_{j,n}|) - \sum_{j=1}^{d} a_j \right| \geq \epsilon \sum_{j=1}^{d} a_j \right)$$

$$\leq 2d \exp(-\frac{n\Delta^2}{2}) + 2\exp\left( -\frac{2n\epsilon^2 \cos^4((1+\Delta)\frac{\pi}{4})}{\pi^2} \right).$$

for any $\{Y_{j,k}\}$ such that $Y_{j,1}, \cdots, Y_{j,n} \overset{\text{iid}}{\sim} \text{Cauchy}(0,1)$ for any $j \in [d]$. Furthermore, $(|Y_{j,1}|, \cdots, |Y_{j,n}|) \overset{d}{=} (\tan(X_{j,1}), \cdots, \tan(X_{j,n}))$ for $X_{j,1}, \cdots, X_{j,n} \overset{\text{iid}}{\sim} \text{Unif}(0, \frac{\pi}{2})$. So we only need to show that

$$P\left( \left| \sum_{j=1}^{d} a_j \text{median}(\tan(X_{j,1}), \cdots, \tan(X_{j,n})) - \sum_{j=1}^{d} a_j \right| \geq \epsilon \sum_{j=1}^{d} a_j \right) \tag{17}$$

$$\leq 2d \exp(-\frac{n\Delta^2}{2}) + 2\exp\left( -\frac{2n\epsilon^2 \cos^4((1+\Delta)\frac{\pi}{4})}{\pi^2} \right).$$

for any $\{X_{j,k}\}$ such that $X_{j,1}, \cdots, X_{j,n} \overset{\text{iid}}{\sim} \text{Unif}(0, \frac{\pi}{2})$ for any $j \in [d]$.

Fix $\Delta \in (0,1)$. We define

$$f(x) = \begin{cases} \tan(x) & \text{if } |x - \frac{\pi}{4}| \leq \Delta\frac{\pi}{4}, \\ \frac{1}{\cos^2((1-\Delta)\frac{\pi}{4})}(x - (1-\Delta)\frac{\pi}{4}) + \tan((1-\Delta)\frac{\pi}{4}) & \text{if } 0 < x < (1-\Delta)\frac{\pi}{4}, \\ \frac{1}{\cos^2((1+\Delta)\frac{\pi}{4})}(x - (1+\Delta)\frac{\pi}{4}) + \tan((1+\Delta)\frac{\pi}{4}) & \text{if } (1+\Delta)\frac{\pi}{4} < x < \frac{\pi}{2}. \end{cases}$$

Then $f(x)$ is a differentiable function on $(0, \frac{\pi}{2})$ that equals to $\tan(x)$ in the middle and is linear on both ends.

We can decompose Equation 17 into

$$P\left( \left| \sum_{j=1}^{d} a_j \text{median}(\tan(X_{j,1}), \cdots, \tan(X_{j,n})) - \sum_{j=1}^{d} a_j \right| > \epsilon \sum_{j=1}^{d} a_j \right)$$

$$= P\left( \left| \sum_{j=1}^{d} a_j \tan(\text{median}(X_{j,1}, \cdots, X_{j,n})) - \sum_{j=1}^{d} a_j \right| > \epsilon \sum_{j=1}^{d} a_j \right)$$

$$\leq P\left( \left| \sum_{j=1}^{d} a_j \tan(\text{median}(X_{j,1}, \cdots, X_{j,n})) - \sum_{j=1}^{d} a_j f(\text{median}(X_{j,1}, \cdots, X_{j,n})) \right| > 0 \right)$$

$$+ P\left( \left| \sum_{j=1}^{d} a_j f(\text{median}(X_{j,1}, \cdots, X_{j,n})) - \sum_{j=1}^{d} a_j \right| > \epsilon \sum_{j=1}^{d} a_j \right).$$

For the first part, we have that

$$P\left( \left| \sum_{j=1}^{d} a_j \tan(\text{median}(X_{j,1}, \cdots, X_{j,n})) - \sum_{j=1}^{d} a_j f(\text{median}(X_{j,1}, \cdots, X_{j,n})) \right| > 0 \right)$$

$$\leq \sum_{j=1}^{d} P\left( |\tan(\text{median}(X_{j,1}, \cdots, X_{j,n})) - f(\text{median}(X_{j,1}, \cdots, X_{j,n}))| > 0 \right)$$

$$\leq \sum_{j=1}^{d} P\left( \left| \text{median}(X_{j,1}, \cdots, X_{j,n}) - \frac{\pi}{4} \right| > \Delta\frac{\pi}{4} \right)$$

$$\leq 2d \exp\left( -\frac{n\Delta^2}{2} \right)$$

where we apply Lemma E.1 in the last step.

For the second part, we first know that $\text{median}(X_{j,1}, \cdots, X_{j,n}) - \frac{\pi}{4}$ is sub-Gaussian variable with $\sigma^2 = \frac{\pi^2}{16n}$ from Lemma E.1. Then $f(\text{median}(X_{j,1}, \cdots, X_{j,n})) - \mathbb{E}f(\text{median}(X_{j,1}, \cdots, X_{j,n}))$ is sub-Gaussian variable with $\sigma^2 = \frac{1}{\cos^4((1+\Delta)\frac{\pi}{4})}\frac{\pi^2}{16n}$ because $f'(x) \le \frac{1}{\cos^2((1+\Delta)\frac{\pi}{4})}$. And $\sum_{j=1}^{d} a_j f(\text{median}(X_{j,1}, \cdots, X_{j,n})) - \sum_{j=1}^{d} a_j \mathbb{E}f(\text{median}(X_{j,1}, \cdots, X_{j,n}))$ is sub-Gaussian variable with $\sigma^2 = \frac{1}{\cos^4((1+\Delta)\frac{\pi}{4})}\frac{\pi^2}{16n}\left(\sum_{j=1}^{d} a_j\right)^2$. When $n \ge \frac{\pi^3}{2\epsilon^2 \cos^4((1+\Delta)\frac{\pi}{4})}$, it holds that

$$
\begin{aligned}
|\mathbb{E}f(\text{median}(X_{j,1}, \cdots, X_{j,n})) - 1| &\le \mathbb{E}\left|f(\text{median}(X_{j,1}, \cdots, X_{j,n})) - f(\frac{\pi}{4})\right| \\
&\le \max_x f'(x)\mathbb{E}\left|\text{median}(X_{j,1}, \cdots, X_{j,n}) - \frac{\pi}{4}\right| \\
&\le \frac{1}{\cos^2((1+\Delta)\frac{\pi}{4})}\sqrt{2\pi}\frac{\pi}{4\sqrt{n}} \le \frac{\epsilon}{2}
\end{aligned}
$$

and

$$
\begin{aligned}
&P\left(\left|\sum_{j=1}^{d} a_j f(\text{median}(X_{j,1}, \cdots, X_{j,n})) - \sum_{j=1}^{d} a_j\right| > \epsilon \sum_{j=1}^{d} a_j\right) \\
\le &P\left(\left|\sum_{j=1}^{d} a_j f(\text{median}(X_{j,1}, \cdots, X_{j,n})) - \sum_{j=1}^{d} a_j \mathbb{E}f(\text{median}(X_{j,1}, \cdots, X_{j,n}))\right| > \frac{\epsilon}{2}\sum_{j=1}^{d} a_j\right) \\
&+ \sum_{j=1}^{d} P\left(|\mathbb{E}f(\text{median}(X_{j,1}, \cdots, X_{j,n})) - 1| > \frac{\epsilon}{2}\right) \\
\le &2\exp\left(-\frac{2n\epsilon^2 \cos^4((1+\Delta)\frac{\pi}{4})}{\pi^2}\right).
\end{aligned}
$$

Combining these two parts, we can get that

$$
\begin{aligned}
&P\left(\left|\sum_{j=1}^{d} a_j \text{median}(\tan(X_{j,1}), \cdots, \tan(X_{j,n})) - \sum_{j=1}^{d} a_j\right| \ge \epsilon \sum_{j=1}^{d} a_j\right) \\
\le &2d\exp(-\frac{n\Delta^2}{2}) + 2\exp\left(-\frac{2n\epsilon^2 \cos^4((1+\Delta)\frac{\pi}{4})}{\pi^2}\right)
\end{aligned}
$$

for $n \ge \frac{\pi^3}{2\epsilon^2 \cos^4((1+\Delta)\frac{\pi}{4})}$. □

### E.4 MORE RESULTS

As mentioned in Section 4.2, we also explore how learning rate can affect the performance of different optimizers by using peak learning rate $3 \times 10^{-4}$ and $1.8 \times 10^{-3}$ for all the adaptive optimizers. The training losses are plotted in Figure 3. All the optimizers perform worse with smaller learning rate, which aligns with the common understanding that optimizers tend to work better with larger learning rate as long as the training is still stable. When a larger learning rate is used, the performance of Adam is improved but the performance of rotated Adam becomes worse than with the default learning rate. The training with AdaSGD even completely failed. This suggests that another advantage of Adam over AdaSGD: it can maintain stable training at a larger learning rate, which is often beneficial to faster and more efficient convergence.

As mentioned in Section 4.3, we tried different batch sizes when training ResNet and the results are in Table 4.

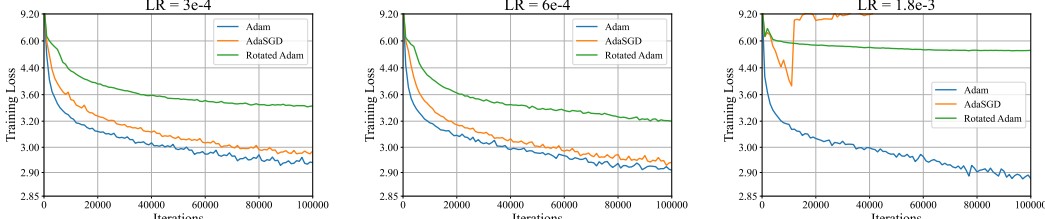

Figure 3: Training losses of `Adam`, `AdaSGD` and rotated `Adam` on GPT-2 under different learning rates. All the optimizers perform worse with smaller learning rate $3 \times 10^{-4}$. Only `Adam` will perform better with larger learning rate $1.8 \times 10^{-3}$.

| Batch Size | SGD | AdaSGD | Adam | Rotated Adam |
|---|---|---|---|---|
| 16 | 0.0777 | 0.114 | 0.064 | 0.0905 |
| 64 | 0.0698 | 0.0854 | 0.0472 | 0.0574 |
| 256 | 0.0723 | 0.0787 | 0.0359 | 0.0485 |
| 1024 | 0.1115 | 0.0915 | 0.0735 | 0.0817 |

Table 4: Training losses of ResNet for different optimizers and different batch sizes within 20 epochs. For each setting, we choose the optimal performance over all the learning rates. The performance of `Adam` is consistently the best among all four optimizers.

