# OpenReview forum: "Adam Exploits $\ell_\infty$-geometry of Loss Landscape via Coordinate-wise Adaptivity"
_ICLR.cc/2025/Conference — ICLR 2025 Spotlight_

### Official Review · Reviewer_chAM · 2024-11-03

**Soundness:** 2
**Presentation:** 2
**Contribution:** 3
**Rating:** 8
**Confidence:** 4

**Summary:**

The authors give a unified proof for AdaSGD, Adam, and blockwise Adam under a new Lipschitz and noise assumption. The new assumption does not only consider the gradient Lipschitz of the full gradient but also the Lipschitz property for each block (for SGD, the full parameter, for Adam, each coordinate).  From the theoretical results, the authors find that (1,1)-norm is the critical value for convergence. Experimental results validate the conclusion for Adam-type algorithms.

**Strengths:**

1.  The authors propose a unified algorithm that contains Adam, AdaSGD, and blockwise Adam.

2. The authors propose a more detailed assumption on gradient Lipschitz to characterize the underlying function carefully. Thus, they can give a tighter bound than giving the overall Lipschitz constant.

3. The authors find that (1,1)-norm of hessian is positively related to the performance of Adam in both theoretical analysis and experimental validation.

**Weaknesses:**

1.  From my point of view the theorem in section 3.3 has already covered the results in section 3.2, making section 3.2 meaningless.

2. The authors claim that in their proof, we can see the reason that Adam can be better than SGD, while the explanation of the results is only given by $\sup_x ||\nabla^2 L(x)||_{1,1} \leq \sup_x ||\nabla^2 L(x)||_2$. It should have some reasonable examples.

3. In Table 1,  since the convergence of AdaSGD is related to 2-norm instead of (1,1)-norm, why do the authors not report the 2-norm instead of reporting (1,1)-norm twice?

4. There are some typos in the paper:

e.g., The first line of notation $\sum_{i=1}^d$ instead of $\sum_{i=1^d}$, $\infty$ instead of $infty$. The name of the algorithm is "Adam-mini" instead of $Adamini$. I do not carefully check every detail of the writing, but the authors should go through and correct the typos.

**Questions:**

1.  Do the results in Section 3.3 cover results in Section 3.2? If so, why should we introduce section 3.2? If not, can you specify the major difference?

2. A simple example: when we optimize a quadratic function with a positive diagonal matrix. In this case, Adam converges much faster than SGD while the (1,1)-norm of the matrix is always larger than the 2-norm of the matrix, because the (1,1)-norm is the sum of diagonal value while the 2-norm is the maximum value of the diagonal entries.  It seems that the result in the simple case contradicts the results provided in section 3.3.

Can the authors give some concrete examples showing the correctness of the claim?

---

> ### Comment · Reviewer_chAM · 2024-11-16
> **Further Question on Theorem 3.12**
>
> For SGD, the convergence speed when $\sigma = 0$ should be in the order of $O(|\nabla ^2 L(x)| _2^ {1/2} T^ {-1/4})$, which seems to be faster than either AdaSGD which is in the
> order of $O(d^{1/2} |\nabla^2 L(x)| _2 ^{1/2} T^{-1/4} )$
> or Adam which is in the order of $O(d ^{1/2} |\nabla ^2 L(x)| _{1,1} ^{1/2} T ^{-1/4})$.
>
> I do think the authors give an important direction that we have to find some rotation variant assumption to show the benefit of Adam, but can the proposed assumption really explain the advance of Adam compared with SGD instead of AdaSGD?

---

> > ### Author Response · Authors · 2024-11-20
> > **Comparison between Adam and SGD**
> >
> > We appreciate the reviewer’s observations and we will directly compare Adam with SGD here. To clarify, the classical analysis for SGD yields the result $\min \\|g_t\\|\_2 \leq O(\\|\nabla^2 L(x)\\|\_2^{1/2} (L(x\_0) - \min L(x))^{1/2} T^{-1/2})$ when the noise $\sigma = 0$. Our results for deterministic Adam establish $\min \\|g_t\\|\_1 \leq O(\\|\nabla^2 L(x)\\|\_{1,1}^{1/2} (L(x\_0) - \min L(x))^{1/2} T^{-1/2})$ (Corollaries 3.7 and 3.8). It is important to note that these results use different metrics on the left-hand side ($\\|g_t\\|\_2$ vs. $\\|g_t\\|\_1$), making a direct comparison between the rates for SGD and Adam inappropriate. However, we can derive comparable bounds below from the relationship $||g_t||_2 \leq ||g_t||_1 \leq \sqrt{d} ||g_t||_2$:
> >
> > - SGD: $\min ||g_t||_1 \leq O(\sqrt{d} ||\nabla^2 L(x)||_2^{1/2} (L(x_0)-\min L(x))^{1/2} T^{-1/2})$
> > - Adam: $\min ||g_t||\_2 \leq O(||\nabla^2 L(x)||_{1,1}^{1/2} (L(x_0)-\min L(x))^{1/2} T^{-1/2})$
> >
> > When comparing the results under the same metric, we can have
> >
> > - For $\\|g_t\\|_2$, the constant for SGD depends on $\\|\nabla^2 L(x)\\|\_2^{1/2}$, while for Adam, it depends on $\\|\nabla^2 L(x)\\|\_{1,1}^{1/2}$. In this sense, Adam’s rate appears worse.
> > - For $\\|g_t\\|_1$, the constant for SGD becomes $\sqrt{d} \\|\nabla^2 L(x)\\|\_2^{1/2}$, and for Adam, it remains $\\|\nabla^2 L(x)\\|\_{1,1}^{1/2}$. Then it is reasonable to compare (1,1)-norm with $d$ times $\ell\_2$ norm as done in our paper.
> >
> > We hope this explanation clarifies the relationship between our results and prior analyses of SGD, addressing the concerns raised.

---

> > > ### Comment · Reviewer_chAM · 2024-11-26
> > >
> > > Thank you for your careful explanation. Is the 2-norm in Theorem 3.12 a typo? If not, then it seems that Theorem 3.12 is not as tight as Theorem 3.5.

---

> > > > ### Author Response · Authors · 2024-11-26
> > > >
> > > > The $\ell_2$ norm is not a typo. As mentioned in line 299-305, theorem 3.5 is a special case of theorem 3.12. When each block only contains one element, $d_b=1$ and $\|\|\bar{\mathbf{g}}\_{t,(b)}\|\|\_2 = |\bar{\mathbf{g}}\_{t,b}|$. Then the LHS of theorem 3.12 becomes $\min_{T/2 < t \leq T} \mathbb{E} \|\| \bar{\mathbf{g}}_t\|\|\_1$, which is the same as the LHS of theorem 3.5.

---

> > > > > ### Comment · Reviewer_chAM · 2024-11-27
> > > > >
> > > > > Thank you for the explanation, I have no further questions and raise my score.

---

> ### Author Response · Authors · 2024-11-20
>
> We sincerely thank the reviewer for their thoughtful feedback on paper writing and for acknowledging the significance of our work in providing a unified framework and insights into Adam-type algorithms. Below, we address the specific concerns raised:
>
> 1. **Relationship between Section 3.2 and Section 3.3.**
>
>     Section 3.2 focuses on the assumptions, convergence rate, and hyperparameter choices for Adam, following a standard structure commonly seen in related work that centers specifically on Adam. We decided to present it as a standalone subsection rather than merging it into Section 3.3 to ensure that researchers interested in Adam can easily compare our results and assumptions with prior analyses. If Adam were only treated as a special case of blockwise Adam in Section 3.3, researchers would need to spend additional effort translating our general results into the specific Adam framework.
>
>     Section 3.3, on the other hand, provides a more general framework that encompasses various optimization algorithms including AdaSGD. This broader approach highlights the distinctions among these algorithms and how these differences impact their convergence rates. By including AdaSGD as a notable special case in Section 3.3, we demonstrate how our general framework subsumes multiple algorithms and contributes to a more unified understanding of their behavior.
>
>     This separation ensures clarity and accessibility for both researchers focused on Adam and those interested in broader generalizations.
>
> 2. **Justification of (1,1)-norm and $\ell_2$ norm.** Please check the general response in which we clarify the typo.
> 3. **Explanation for table 1.** You are right that the convergence of AdaSGD is related to 2-norm so we report that 2-norm (spectral norm) is 1 in the caption. Leaving the (1,1)-norm cell for AdaSGD blank might have disrupted the presentation of the table, so we chose to include it for completeness even though it is not necessary for interpreting its convergence behavior. We appreciate this observation and will clarify this in the revised manuscript to avoid any misunderstanding.

---

### Official Review · Reviewer_3yjk · 2024-11-03

**Soundness:** 3
**Presentation:** 3
**Contribution:** 3
**Rating:** 8
**Confidence:** 4

**Summary:**

This paper presents a new theoretical analysis of the Adam optimizer, highlighting its advantages over vanilla SGD in training deep neural networks like GPT-2 and ResNets. By relying on loss smoothness under $l\infty$-norm geometry instead of the more common $l2$-norm smoothness assumption, the authors argue that Adam's success over SGD is due to its coordinate-wise adaptivity which allows to exploit some properties of the presented models.The analysis is also extended to blockwise Adam which is a generalized form of Adam.

The proposed theory is empirically verified Adam on rotated and unrotated loss landscapes, verifying the claim that Adam uses non-rotation-invariant features.

**Strengths:**

1. Even if the paper could be even more polished (see Question 1 for a comprehensive list of needed corrections), the paper is overall very-well written and interesting. I carefully read the main text and the appendices and found the proofs very clear and did not find mathematical errors.

2. The authors introduce what seems to me is a novel framework to better capture Adam's coordinate-wise adaptivity, namely the $l\infty$ geometry that allows them to get tighter convergence bounds than previous work (Défossez et al., 2022) that used $l2$ smoothness of the objective function.

3. The authors demonstrate Adam's sensitivity to rotation. This is used to show that Adam overperforms SGD thanks to non-rotation-invariant properties.

4. Blockwise-Adam is proposed allowing adaptive updates across parameter groups, which could be used for large models.

5. An empirical validation on models like GPT-2 and ResNet-18 is provided, which supports the theoretical findings. The results on rotated Adam are in my opinion particularly interesting and the Appendix D.1 answers a natural question that arises when reading the paper as to how to apply an orthogonal rotation on the parameters on large models.

6. The provided insights could be used to further improve adaptive methods in deep learning and maybe even design adaptive methods "adapted" to specific models as  the discussed coordinate-wise adaptivity is probably heavily linked to specific properties of the studied models.

**Weaknesses:**

1. The convergence rate improvements seem a bit incremental when compared to (Défossez et al., 2022) and might not translate into practical gains.

2. The use of non-standard $l\infty$ smoothness assumption may limit the generalizability of the proposed results.

3. The empirical analysis of rotation sensitivity is very interesting but a bit limited in scope. The effort of including a ResNet-18 to explore a different kind of architecture is commendable but a more diverse set of architectures would be very interesting to study. It might provide deeper insight into why Adam is impacted by certain rotations and also in which cases do the non-rotation-invariant properties of Adam easily emerge.

4. The blockwise Adam variant is interesting but under-exploited as there is no practical guidance on choosing parameter blocks, even though the related works of Adamini (Zhang et al., 2024b) and Adalayer (Zhao et al., 2024) are mentioned.

5. The high memory consumption of Adam is only mentioned in the introduction. In practice, Adam's higher memory usage when compared to SGD (three times as much) is a real drawback, particularly for large models relying on the Transformer architecture which is studied in this work (GPT-2). One could ask if training a three times bigger model with SGD before reducing its size with common techniques (distillation, pruning) would lead to better results.

6. The focus on the case $\beta_1=0$ seems a bit far-fetched when looking at the results of Table 1 as the final losses obtained are far better with ($\beta_1 = 0.9, \beta_2 = 0.99$)

**Questions:**

1. A thorough proofread is needed as some typos/redundancies exist even though the paper is quite polished (eg. l.49 "If Adam optimizes much slower more slowly after" -> "If Adam optimizes much slower" or "If Adam optimizes more slowly"; l.102: "for $p \in [1, infty]$ " -> "for $p \in [1, \infty]$"; l. 459: "OpenWebText corups"->"OpenWebText corpus".

2. A naive question one could ask is what would the results of rotated Adam be when compared to Adam on standard CNNs/MLPs and recurrent neural networks? What could be the intuition behind that?

3. Are there any specific patterns in the rotations that worsen performance, or does any random rotation lead to degradation? Could the pattern be extracted from the data/model? A more detailed work on the rotation sensitivity would be very interesting.

---

> ### Author Response · Authors · 2024-11-20
>
> We sincerely thank the reviewer for recognizing the novelty and significance of our results and for highlighting several insightful directions to further enhance the paper. We greatly appreciate the reviewer’s constructive suggestions for expanding our work. However, due to limited time and computational resources, we were unable to conduct all the extensive experiments, such as testing more model architectures or exploring different kinds of rotations. We hope that our theoretical findings and preliminary empirical results can guide future work and inspire further exploration by the community. Below, we address the specific issues:
>
> 1. **Understanding convergence result.** Our convergence rate for Adam improves the dependence on the number of parameters, $d$, by replacing the constant $d$  times $\ell_2$-norm of the Hessian with the (1,1)-norm of the Hessian. For large models, where $d$ can be much larger than the total number of training steps $T$, this refinement results in a more practical and meaningful bound. Additionally, we argue that the previous $\ell_2$-norm-based bounds for Adam can be vacuous, as they do not distinguish between standard Adam and rotated Adam. Since rotated Adam has significantly worse performance in practice, the $\ell_2$-norm-based analysis does not fully capture Adam’s behavior. Our results address this limitation by using a more appropriate norm for Adam’s coordinate-wise adaptivity.
> 2. **Smoothness assumption.** As mentioned above and in our introduction (line 45-51, 366-373), we believe the standard $\ell_2$ norm is inadequate for tight analysis of Adam. The benefit of coordinate-wise adaptivity can only be captured under a non-standard $\ell_\infty$ norm geometry, which justifies the need for a new smoothness assumption. While this assumption may limit the direct comparability with other optimization methods, it provides a more accurate framework for understanding Adam’s behavior.
> 3. **Larger model with SGD vs smaller model with Adam.** Our analysis focuses on comparing Adam and SGD in terms of the number of iterations for the same problem. Comparing models of different sizes involves more complex issues related to expressiveness and scaling laws, which are outside the scope of our work. Nevertheless, we agree that exploring such trade-offs could be an interesting direction for future work.
> 4. **General $\beta_1$.** Please refer to the beginning of Section 3 (lines 151–157).
>
>     > Similar to previous work (Défossez et al., 2022), our analysis could be extended to the most general case of Adam, where both $\beta_1$, $\beta_2$ are non-zero, but the rate becomes strictly worse than the RMSProp (the case of $\beta_1=0$), as there will be some extra polynomials of $\frac{1}{1-\beta_1}$.
>     We decide not to include the result for the most general case, on one hand for ease of presentation, and on the other hand, because such result can not explain the optimization benefit of momentum ($\beta_1>0$) in practice and does not add any insight on the benefit of Adam. We hypothesize that we are missing some important features of loss landscape of transformers in the theoretical assumptions and we leave this for future work.
>     >

---

> > ### Comment · Reviewer_3yjk · 2024-12-02
> >
> > Thank you for your answers.
> > I would be really interested to see more architectural setups integrated in your experimental framework (with simple architectures such as MLPs or CNNs), to see if the analysis provided can be extended.
> >
> > A study on the rotation sensitivity would also further strengthen the paper, although it might be a full study by itself.
> >
> > I raise my score to 8 as I think this is a good paper.

---

### Official Review · Reviewer_GxXu · 2024-11-09

**Soundness:** 3
**Presentation:** 3
**Contribution:** 3
**Rating:** 6
**Confidence:** 4

**Summary:**

The paper studies the superior performance of Adam compared to SGD from a theoretical perspective.

Specifically, existing analysis on Adam is commonly performed under the $\ell_2$ norm. This paper argues that Adam's advantages arise from its sensitivity to the $\ell_\infty$ geometry of the loss landscape, rather than the $\ell_2$ geometry. By introducing $\ell_\infty$ smoothness measures, the authors develop a new theoretical framework to explain Adam's faster convergence rates. The authors further extend this framework to cover blockwise Adam variants such as Adalayer and Adam-mini.

To support the theoretical findings, the paper presents experimental evidence that:

1. Adam performs poorly on rotated loss landscapes, demonstrating its dependence on non-rotation-invariant properties.

2. Adam outperforms SGD in cases where the $\ell_\infty$ smoothness measure is significantly smaller than the $\ell_2$ smoothness measure, demonstrating that $\ell_\infty$ measures are more adequate to characterize Adam's performance.

**Strengths:**

1. The paper draws an insight that Adam is permutation-invariant, but not rotation-invariant is crucial, while SGD is rotation-invariant. I believe this property is highly related to the performance difference between Adam and SGD.

2. Based on $\ell_\infty$ smoothness measures, the paper provides a general framework to analyze Adam and its blockwise variants such as Adam-mini and Adalayer. This contribution is timely, given the increasing interest in blockwise optimization approaches aimed at reducing memory overhead in training large language models. The framework of this paper can potentially guide the design of new Adam variants.

3. Experiments show that different $\ell_\infty$ smoothness measures indeed lead to different performance of Adam. This provide a strong evidence of the theoretical findings.

**Weaknesses:**

While the paper provides good insights into Adam's sensitivity to $\ell_\infty$ geometry, the proposed theorems using the $ ||\cdot ||_{1,1}$ norm may not fully capture this sensitivity, particularly in explaining the performance gap between SGD and Adam. Two specific concerns are as follows:

- For convex problems with a positive semi-definite Hessian $B$, it holds that:

$$   ||B||_{1,1} \geq \mathrm{trace}(B) \geq || B ||_2 $$

Thus, for a wide class of problems, we have $ \sup_x ||B||2 $ smaller than $\sup_x ||B||_{1,1}$. However, the authors claim ``the latter is typically much smaller when Adam optimizes faster.''  This assertion seems counterintuitive given the inequality. It requires further justification in practical neural network training scenarios.


- In the quadratic loss experiments (Section 4.1), Table 1 shows that Adam still outperforms both SGD and AdaSGD, even for quadratic cases $\sup_x ||\nabla^2 L(x)||_{1,1}$ is larger than $\sup_x ||\nabla^2 L(x)||_2$. This result appears inconsistent with the authors' theoretical argument, as pointed out in the last bullet point.

**Questions:**

1. Zhang et al. (2024a) also characterize the advantage of Adam to SGD. They have a theory on quadratic problems. Can the authors compare the results to that of Zhang et al. (2024a)?

2. The authors point out that rotating the loss can easily hamper the performance of Adam. However, in practice, Adam performs on par or better than SGD for most neural networks. This means that deep neural networks are not ``randomly rotated''. Why? Can the authors discuss this?

---

> ### Author Response · Authors · 2024-11-20
>
> We sincerely thank the reviewer for acknowledging the importance of the rotation invariance property towards understanding gap between Adam and SGD and the value of our results in guiding the design of new optimization algorithms. We greatly appreciate the constructive feedback and will address the concerns in detail below.
>
> 1. **Justification of (1,1)-norm and $\ell_2$ norm.** See general response in which we clarify the typo.
> 2. **Comparison with Zhang et al. (2024a).**
>
>     Both our work and Zhang et al. (2024a) aim to highlight Adam’s advantages over SGD under specific properties of the Hessian matrix. At a high level, our conclusions align, as $\\| \nabla^2 L(x) \\|_{1,1}$ is typically smaller than $d \\| \nabla^2 L(x) \\|_2$ when there is block heterogeneity in the Hessian matrix $\nabla^2 L(x)$. The quadratic loss example in Section 4.1 of our work serves as an extreme example of such block heterogeneity, supporting this shared insight.
>
>     There is difference in terms of the specific results. The theoretical result in Zhang et al. (2024a) is for strongly convex quadratic loss functions and only holds for the degenerate case $\beta_2 = 1$ in Adam. Moreover, as shown in  Zhang et al. (2024a), $\beta_2=1$ is crucial in their analysis because any $\beta_2<1$ will lead to non-convergence of Adam. We believe this is not a very realistic regime for Adam analysis because, in deep learning practice, $\beta_2$ is typically chosen to be strictly smaller than 1. Setting $\beta_2=1$ leads to suboptimal performance because the denominator in the update rule of Adam becomes fixed at its initialization, causing Adam to degenerate into SGD with constant diagonal preconditioner. In contrast, our analysis focuses on the convergence of Adam in the standard non-convex stochastic optimization setting and is valid for a wide range of $\beta_2$ values.
>
> 3. **Non-rotation-invariance of neural networks.** We appreciate this insightful question. We believe theoretically understanding the advantage of the standard coordinate system is an important and interesting future direction while we will discuss our initial thoughts on it below. The lack of rotation invariance in deep neural networks can be attributed to both architecture and data:
>
>     **Architecture**:  The hierarchical structure and the properties of specific components like activation functions and normalization layers can cause non-rotation-invariance.  For example, coordinate-wise nonlinear activation functions, such as ReLU, introduce dependencies that are not preserved under rotation. These elements are tightly coupled to the original coordinate system, and arbitrary rotations can disrupt this alignment.
>
>     **Data**: Specific data formats may also contribute to non-rotation-invariance. For example, in language modeling, each row in the embedding matrix corresponds to different tokens so the embedding matrix itself is not randomly rotated. The imbalance in training data distribution may also attribute to the asymmetry of model parameters and cause Adam to optimize faster than SGD as observed by Kunstner et al. 2024. This explanation is consistent with the quadratic example we provide in Sec 4.1, where different diagonal values corresponds to the frequency of different classes. Such imbalance only allows faster convergence of Adam over SGD in the standard coordinate system, and the optimization advantage disappears when we apply Adam in the rotated coordinate system.
>
> We kindly ask the reviewer to consider revisiting their score in light of our responses, as we believe we have addressed the main concerns about the relationship between different norms. We also welcome the reviewer to raise any additional questions or suggestions to help us further improve our work.
>
> Kunstner, Frederik, et al. "Heavy-tailed class imbalance and why adam outperforms gradient descent on language models." *arXiv preprint arXiv:2402.19449* (2024).

---

> > ### Comment · Reviewer_GxXu · 2024-12-02
> >
> > Thank you for the detailed response. Most of my comments are addressed. I have raised my score.
> >
> > However, as I read the discussion under review chAM, I am confused about why Adam converges faster in the sense of L1 norm of $g_t$, but is slower in the sense of L2 norm of $g_t$. Which bound is tighter? Or both are tight, but the L1 and L2 norms of Adam updates indeed differ in practice?

---

> > > ### Author Response · Authors · 2024-12-03
> > >
> > > The l2 norm bound of gradient for Adam is derived from the inequality $\|g_t\|_2 \leq \|g_t\|_1$, which could be very crude and not necessarily tight (note $\|g_t\|_2 =\|g_t\|_1$ when $g_t$ is 1-sparse). We do not have matching lower bound for that. Also it is theoretically possible that for a specific loss function, Adam finds points with smaller l1 gradient norm than that of SGD, but its l2 gradient norm is larger.
> > >
> > > However, our main point is that to theoretically understand the empirical benefit of Adam over SGD, we have to resort to non-rotational invariant measure and metrics. In this paper we show that $\ell_\infty$ smoothness and $\ell_1$ norm of gradients are good candidates in the sense that Adam can achieve much better rates for $\ell_1$ norm of gradients than SGD and the rotated version of Adam, thanks to the small $\| \nabla ^2 L\| _ {1,1}$ in standard Adam training. We cannot directly analyze the suboptimality of loss due to the non-convexity setting in this paper, but it should be possible to extend the current framework to convex settings so we can properly compare the loss.
> > >
> > > We would greatly appreciate your kind consideration in the evaluation if the above response addresses your concerns.

---

### Author Response · Authors · 2024-11-14
**Clarification of typos in Section 3.3**

We apologize for the significant typo in section 3.3 that seems to cause reviewer GxXu and chAM to misunderstand our results. In line 327, $H(L, i \mapsto 1)$ should be equal to $d \sup_{x \in \mathbb{R}^d} || \nabla^2 L(x) ||\_2$ where $d$ is the number of parameters. And all the $\sup_{x \in \mathbb{R}^d} ||\nabla^2 L(x)||\_2$ in the discussion in section 3.3 should also be replaced by $d \sup_{x \in \mathbb{R}^d} ||\nabla^2 L(x)||\_2$.  **Please** **refer to the updated paper in which we highlight the corrections in red color** and fix other minor typos. This was a purely typographical error, as the correct statement appears in the introduction (lines 77-80) and in the experiments section (Table 1 caption and lines 492-494). We always compare (1,1)-norm divided by d with $\ell_2$ norm in the experiment section. This is consistent because both values are scaled down by  $d$ , allowing for a more interpretable comparison given that  $d$  can be very large, while the $\ell\_2$ norm of Hessian matrix is usually $O(1)$.

We explain why $H(L, i \mapsto 1)$ is $d \sup_{x \in \mathbb{R}^d} ||\nabla^2 L(x)||\_2$ here while we also apologize that we only define $d_b$ in appendix C.1 instead of in the main text. For a partition function $\Phi: [d] \rightarrow [B]$ and any block $b \in [B]$, $d_b$ is the number of parameters that is partitioned into block b, i.e., $d_b = | \\{i \in [d]| \Phi(i)=b \\}|$. For $\Phi\_{\mathtt{AdaSGD}}: i \mapsto 1$ , $||x||\_{\Phi\_{\mathtt{AdaSGD}}} = \frac{||x||_2}{\sqrt{d}}$ and the definition of 3.10 becomes $\sqrt{d} ||\nabla L(x)-\nabla L(y)||\_2 \leq H_1 \frac{||x-y||\_2}{\sqrt{d}}$. Therefore, $H(L, i \mapsto 1) = H_1 = d \sup\_{x, y \in \mathbb{R}^d} \frac{||\nabla L(x)-\nabla L(y)||\_2}{||x-y||\_2}=d \sup\_{x \in \mathbb{R}^d} ||\nabla^2 L(x)||\_2$.

As pointed out by reviewer GxXu and chAM, $\ell_2$ norm itself is indeed always smaller than (1,1)-norm. Instead, it is reasonable to compare (1,1)-norm with d times $\ell_2$ norm, either of which can be larger for some matrices. That’s why we propose the simple quadratic function example in Section 4.1 to facilitate understanding. When the Hessian is diagonal, and diagonal elements decay quickly, (1,1)-norm is much smaller than d times $\ell_2$ norm and Adam optimizes faster than AdaSGD. When the Hessian no longer exhibits such sparse or block heterogeneity structure, (1,1)-norm is larger than d times $\ell_2$ norm and Adam optimizes more slowly than AdaSGD. This experiment shows the high correlation between optimization speed and corresponding smoothness, supporting the insight provided by Theorem 3.12.

---

### Author Response · Authors · 2024-11-25
**Follow-up on discussion**

Thank you once again for your thoughtful and constructive comments on our paper. We hope our responses have addressed the concerns you raised and provided additional clarity on the key points of our work.

As the discussion period is approaching its conclusion, we want to kindly follow up to check if there are any further questions or aspects you would like to discuss. Additionally, we were wondering if our clarifications have adequately addressed your concerns and if they might lead you to reconsider your evaluation of the paper.

We deeply appreciate the time and effort you have dedicated to reviewing our submission and would be grateful for any additional thoughts you may have.

---

### Meta-Review · Area_Chair_WafK · 2024-12-20

**Metareview:**

This paper analyzes the superior performance of Adam over SGD, arguing that Adam's advantage stems from its better sensitivity to the smoothness of the loss landscape and providing a new convergence analysis under a novel $\ell_\infty$ smoothness assumption, and extend this to a blockwise Adam variant. The reviewers noted certain weaknesses, including incremental improvements, limited generalizability, lack of practical guidance on blockwise Adam, but they are largely outweighed by its strengths, such as a novel framework for understanding Adam's adaptivity, strong theoretical and empirical analysis demonstrating its rotation sensitivity, and a unified algorithm encompassing several Adam variants, ultimately offering valuable insights that justify acceptance.

**Additional Comments On Reviewer Discussion:**

The reviewer discussion addressed several points, including comparisons to prior work, the non-rotational-invariance of neural networks, and a few other minor issues. The authors addressed the concerns by clarifying the justification of (1,1)-norm, differentiating its analysis from prior work (Zhang et al., 2024a) by highlighting its broader applicability to realistic Adam settings and non-convex optimization, and providing insightful reasoning for the non-rotation-invariance of neural networks due to both architecture and data characteristics. These responses as well as the clarification about typos in Section 3.3 seemed to address most reviewer concerns.

---

### Decision · Program_Chairs · 2025-01-22

Accept (Spotlight)